



# Predicting Water Retention Curves for Binary Mixtures - Concept and Application for Constructed Technosols

Moreen Willaredt[1], Andre Peters[2], and Thomas Nehls[1]

[1]Department of Ecohydrology & Landscape Evaluation, Institute of Ecology, Technische Universität Berlin, Ernst-Reuter-Platz 1, 10587 Berlin
[2]Department for Soil Science and Soil Physics, Institute of Geoecology, Technische Universität Braunschweig, Langer Kamp 19c, 38106 Braunschweig

**Correspondence:** Moreen Willaredt (moreen.willaredt@campus.tu-berlin.de)

**Abstract.** Constructed Technosols are important means to substitute natural soil material such as peat and geogenic material to be used in urban green infrastructure. One of the most important features of such soils is related to the water cycle and can be described by the soil water retention curve (WRC). The WRC depends on the composition of the constructed Technosols e.g. their components and their mixing ratio. The diversity of possible components and the infinite number of mixing ratios

practically prohibit the experimental identification of the optimal composition regarding the targeted soil functions. In this study we propose a compositional model for predicting the WRC of any binary mixture based on the measured WRCs of it's two pure components only (basic scheme) or with one additional mixture (extended scheme). The model is developed from existing methods for estimating the porosity in binary mixtures. The compositional model approach was tested for four data sets of measured WRCs for different binary mixtures taken from the literature. To assess the suitability of these mixtures for

typical urban applications, the distribution of water and air in $50\,\mathrm{cm}$ high containers filled with the mixtures was predicted under hydrostatic conditions. The difference between the maxima of the pore-size distributions $\Delta\mathrm{PSD_{max}}$ of the components indicates the applicability of the compositional approach. For binary mixtures with small $\Delta\mathrm{PSD_{max}}$, the water content deviations between the predicted and the measured WRCs range from $0.004$ to $0.039\,\mathrm{m^3\,m^{-3}}$. For mixtures with a large $\Delta\mathrm{PSD_{max}}$, the compositional model is not applicable. The knowledge of the WRC of any mixing ratio enables the quick choice of a

composition, which suits the targeted application.

## 1   Introduction

Due to soil sealing the natural soil functions that are involved in regulating water cycles and the energy balance in urban environments are severely disturbed. Therefore, urban problems like pluvial flooding or the intensification of the urban heat island effect are challenging the health and quality of living in urban areas. Climate change intensifies these urgent problems.

In fact, plants and their substrates, in the form of green roofs (Molineux et al., 2009; Eksi et al., 2020), facade greening, urban trees pits (Vidal-Beaudet et al., 2018; Yilmaz et al., 2018), and ornamental raised beds (Pitton et al., 2022), can increase the resilience towards extreme weather events when they are re-introduced to sealed urban areas. The effectiveness of secondary urban greening (Nehls et al., 2015) is dependent upon its brown infrastructure parts (Pouyat et al., 2010). Constructed



Technosols, soil-like substrates or growing media restitute the functions of the former unsealed soils at the site. This can be
described as functional de-sealing. The implementation of urban green infrastructure (UGI) on top of sealed soils poses an
increasing demand for soil, planting substrates and constructed Technosols. These constructed Technosols can be engineered
from locally available valuable mineral and organic waste. This is considered a sustainable path to meet the demand (Prado
et al., 2020; Deeb et al., 2020; Fabbri et al., 2021), as it reuses materials that would otherwise be land filled and decreases
the degradation of fertile natural soil resources and other geogenic materials (Willaredt and Nehls, 2021). Tams et al. (2022)
showed in a life cycle analysis, that the use of recycled brick particles instead of expanded clay reduces the $CO2$ footprint of
the substrate layer by $50\%$ in an extensive green roof. The composition of waste materials and the processing (Ulrich et al.,
2021) are the most important design levers to manipulate the properties according to their targeted application (Rokia et al.,
2014; Fields et al., 2018; Willaredt and Nehls, 2021). Most UGI addresses the re-establishment of soil function related to the
regulation of the water cycle (Grabowski et al., 2022). For that target understanding the functional relationship between the hy-
draulic properties of Technosols and their composition is the prerequisite for formulating purpose-oriented Technosol recipes.
Rokia et al. (2014) were the first to describe the properties of binary and ternary combinations of Technosol components as
functions of their mixing ratio and the employed waste type. Using dose-response curves they were able to describe six basic
soil properties, which are important for agricultural use: $C_{tot}$, $P_{Olsen}$, CEC, $pH_{Water}$, $WC_{-102\,cm}$ and BD. They showed that only
mixtures containing both waste types, mineral and organic, will feature soil-like agronomic properties. Water retention charac-
teristics, distributions of water and air for different energy statuses of water in the soil, determine the successful application of
constructed Technsolos in UGI (Al Naddaf et al., 2011; Caron et al., 2015). Measurements of soil-like, but still unconventional
and unknown components and their combinations require following a protocol guaranteeing reproducibility of the mixture for-
mulation and comparability between the mixtures (Hill et al., 2019; Willaredt and Nehls, 2021). The extensive labor involved
and the demand for cost-intensive equipment limits measurement initiatives that cover the variety of components for Technosol
construction and their infinite possible mixing ratios. Therefore, this study aims to develop a concept that allows the prediction
of WRC based on the measured WRCs of only the constitutes. Concepts that approach soils as (binary) mixtures can be found
in research on soil physical properties after soil amelioration (Abel et al., 2013; Walczak et al., 2002) and in research on soils
containing stones or gravel (Naseri et al., 2019; Zhang et al., 2011). The impact of mixing on soil physical properties, mainly
porosity and saturated hydraulic conductivity, were most comprehensively described for mixtures of coarse and fine particles
with a pronounced particle size difference (Sakaki and Smits, 2015; Zhang et al., 2011; Clarke, 1979). For the porosity in such
mixtures the functional relationship to the composition of the mixture has been described by the delineating concepts „ideal
mixing" and „zero mixing". According to Clarke (1979), binary mixtures that are „ideally mixed" can be distinguished in
two categories depending on their mixing ratio: fine controlled or coarse controlled mixtures. In fine controlled mixtures the
fine component of the mixture determines the properties, and the coarse particles – having no inner porosity - basically reduce
the total volume of the fine component and thus its pores in the mixture by their own volume. In coarse controlled mixtures
the share of fine particles arranges within the pores between coarse particles. In mixtures where the particles are practically
not mixed, „zero mixing", the resulting porosity can be linearly interpolated between the components' porosity. The effect of
the volumetric stone content in fine controlled mixtures on the resulting porosity as well as on the water retention curve and





unsaturated hydraulic conductivity has been successfully described by scaling approaches e.g. from Bouwer and Rice (1984)

and Flint and Childs (1984). With high resolution WRC measurements Naseri et al. (2019) confirm the applicability of the scaling approaches for stony soils with volumetric stone contents not bigger than the order of magnitude of 30 vol%, hence fine controlled mixtures. Sakaki and Smits (2015) measured, in addition to the porosity, the WRCs in mixtures with pronounced particle size difference and found the patterns of „ideal mixing" also reflected in the WRCs. The focus on mixtures with components having distinct particle size differences is a major limitation for the transferability of the prediction concepts to

Technosols. They are mixtures of practice-oriented components with overlapping particle size and pore size distributions e.g. organic and mineral components that present fine graded particle size distributions instead. Therefore, the particles of these components are less likely to arrange within the pore spaces of each other. Hence, the impact of mixing the components on the resulting water retention curves is more likely to be represented by the „zero mixing" concept introduced above.

The purpose of this study is to develop an approach similar to that described for porosity in binary mixtures of coarse and

fine particles, for water retention curves of materials which are suitable for Technosol construction. This is part of the goal to enable the prediction of water retention curves from Technosols formulated as binary mixtures in any mixing ratio based on only a few necessary measurements. We therefore: i) formulate and use a simple compositional model approach to predict the water retention curves of binary mixtures that cover a full range of mixing ratios (from 0/100 to 100/0 (vol/vol) based on the WRCs of the pure components, ii) assess the approach with sets of WRCs of binary mixtures found in the literature for soil-like

components and technogenic components, and iii) present the applicability of the compositional model to predict hydrostatic distribution of water and air using the constructed Technsols as planting substrates in a container.

## 2 Material & Methods

### 2.1 Theory

#### 2.1.1 Pore size distribution in mixture components

Contrary to simple binary mixtures of coarse and fine particles, constructed Technsols are often composed by materials with fine graded particles, which result in rather wide pore size distributions (PSDs). In this study the difference between the maxima of the PSD of both components $\Delta PSD_{max}$ is used as a measure to qualitatively evaluate their similarity. It can be calculated as the difference between the logarithms of the effective radii $R_{eff}$ [m] and $r_{eff}$ [m] at the PSD maxima for the components with larger and smaller components respectively:

$$\Delta PSD_{max} = \log_{10}(R_{eff}) - \log_{10}(r_{eff}) \tag{1}$$

Figure 1 visualizes schematically the proportions of pore radii present in two components and the resulting pore system arrangement with a large $\Delta PSD_{max}$ (Fig. 1 a) and a small $\Delta PSD_{max}$ (Fig. 1 b).





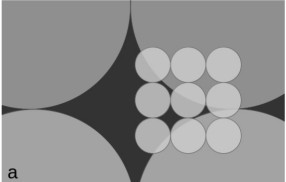
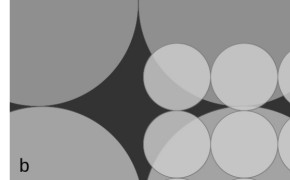

**Figure 1.** Schematic representation of two pore systems a) presenting a pronounced difference in effective pore radii found in two soil components: The pores of the component characterized by smaller pores can arrange within the pores of the component characterized by large pores („ideal mixing") and b) presenting a smaller difference in pore size radii: The pores formed by the particles in the components characterized by the small pore radius do not easily arrange within the larger pore system but rather exist next to each other („zero mixing").

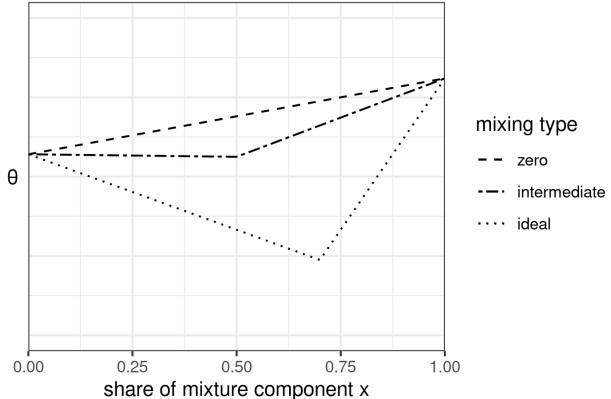

**Figure 2.** Mixing types of water retention characteristics in binary mixtures. (adapted from concept for porosity in binary mixtures, illustrated in Zhang et al. (2011))

### 2.1.2 Adapted Clarke model

The „ideal mixing" approach by Clarke (1979) was formulated to define the lower bound of the resulting porosity in binary

mixtures of fine and coarse particles. As described in the introduction, this approach distinguishes two cases, that depend on the volumetric share of the fine particles in the mixture. As this approach was developed to describe natural soil containing stones or gravel. The volumetric composition describes the volumetric stone content in the mixture. For fine controlled mixtures this implies that the volume of the coarse fraction refers to the solid volume of the contained stones in a background bulk volume of the fine fraction.





When adapting the „ideal mixing" approach to predict the complete water retention curves for any volumetric composition, we refer to the bulk volumes of the components that form the composition. Here $x_i$ refers to the bulk volumetric share that constitutes the mixture (Fig. 2).

$$\theta_{\mathrm{pred}} = \begin{cases} (x_f + \phi_c\, x_c) \cdot \theta_f, & \text{if } x_f \geq x_{\mathrm{crit}}. \\ \left(\phi_c - \frac{x_f(1-\phi_f)}{\phi_c}\right) \cdot \theta_c + \phi_f x_f \theta_f, & \text{otherwise.} \end{cases} \qquad (2)$$

where $\theta_{\mathrm{pred}}$ [-] is the predicted volumetric water content in a mixture and $\theta_f$ [-] and $\theta_c$ [-] stand for the volumetric water
content in the fine and coarse components of the mixture respectively. $\phi_f$ [-] represents the porosity in the fine component and $x_f$ [-] the volumetric share of the fine component *for* the mixture. In fine controlled mixtures ($x_f \geq x_{\mathrm{crit}}$) the porosity that is contained in the bulk volumetric content of the coarse fraction for the mixture $\phi_c \cdot x_c$ [-] will be replaced with fine material. Therefore, the effective share of the fine fraction in the mixture increases accordingly. The volume taken up by the coarse fraction does not contribute to water retention. This corresponds to the scaling approaches tested and approved by Naseri et al.
(2019). In coarse controlled mixtures the porosity in the coarse share of the mixture is reduced by the solid volume introduced by the fine share. However, the water that can be contained within the pores of the fine shares adds to the predicted water content. In binary mixtures with pore systems that are characterized by small $\Delta\mathrm{PSD}_{\mathrm{max}}$ the particles and the pore system formed between them are not going to interlock in a similar way. Hence, they rather exist next to each other and their porosity and correspondingly the water that can be contained in their pore space can be based on the „zero mixing" approach (Fig. 2).

**2.1.3   Compositional model - Basic scheme CM1**

CM1 requires only the water retention curves of the components and their mixing ratio. This approach is a weighted superposition of the WRCs of the two components to predict the WRC of the mixture:

$$\theta_{\mathrm{pred}} = x_a\, \theta_a + (1 - x_a)\, \theta_b \qquad (3)$$

where $x_a$ [-] and $x_b$ [-] represent the bulk volumetric share of component a and b for the mixture, with $x_a + x_b = 1$, $\theta_a$ [-]
and $\theta_b$ [-] are the volumetric water contents at any matric potential of the two single components and $\theta_{\mathrm{pred}}$ [-] is the resulting volumetric water content of the mixture at any matric potential.



### 2.1.4 Compositional model - Extended scheme CM2

The motivation behind the extended scheme is to analyze if a slight increase in measurement effort leads to more sound predictions. CM2 additionally requires the measured WRC of an intermediate mixture containing approximately similar shares

of both components a and b (intermediate mixing type in Fig. 2):

$$\theta_{\text{pred}} = \begin{cases} \frac{x_a}{x_m}\theta_i + \left(1 - \frac{x_a}{x_i}\right)\theta_b, & \text{if } x_a < x_m, \\ \frac{1-x_a}{1-x_i}\theta_m + \left(1 - \frac{1-x_a}{1-x_i}\right)\theta_a, & \text{if } x_a > x_m, \end{cases} \tag{4}$$

where $x_m$ [-] represents the bulk volumetric share of component a in the intermediate mixture and $\theta_m$ [-] the water content in the intermediate mixture. This approach is based on typical calculations for dilution concentrations.

## 2.2 Data sets of binary mixtures

We used four different data sets of WRCs of binary mixtures, covering volumetric mixtures ranging from the pure first component (100/0) to the pure second component (0/100) (Table 1). Three of them represent binary mixtures of one organic and one mineral component mimicking soils and providing soil functions (Walczak et al., 2002; Deeb et al., 2016; Willaredt and Nehls, 2021). The fourth data set (Sakaki and Smits, 2015) represents a mixture of sands with pronounced difference in particle sizes (Fig. 3). The data of Walczak et al. (2002) was digitally extracted from their graphs using the open access software

Engauge-digitizer 12.1 (Mark Mitchell and et al, 2019). The other three data sets were available as raw data.

## 2.3 Mixture preparation and WRC measurement

Deeb et al. (2016) combined excavated deep soil material from construction activity (EDH) with green waste compost (GWC) to the mixtures containing volumetric share of 0, 10, 20, 30, 40, 50 and 100 % (vol), denominated C0E10, C1E9, C2E8, C3E7 C4E6, C5E5 and C10E0 respectively. Four replicates of each mixture were implemented in planting containers. Samples were

taken from their surface. The volumetric water contents of the samples were assessed at 8 matric potentials using the sand box method for matric potentials h of -2, -9.8 and $-31\,\text{cm}$ and a pressure-plate apparatus for the matric potentials h of -310, -980, -1550, -4910 and $-15\,540\,\text{cm}$. Walczak et al. (2002) composed mixtures of peat and sand. They combined these components to mass specific ratios (dry peat mass) of 0, 5, 20, 40, 60, 80 and 100 % (mass). For our analysis the volumetric peat content $x_{i,v}$ of each mixture was determined based on the given bulk densities ($BD_{\text{meas}}$) of the mixtures using the following equation:

$x_{i,v} = x_{i,m} \cdot \frac{BD_{\text{meas}}}{BD_{\text{peat}}}$. The BD of peat and sand are $0.33\,\text{g cm}^{-3}$ and $1.86\,\text{g cm}^{-3}$, respectively. Table 1 summarizes the volumetric ratios of the mixtures and the deviations between the measured and calculated BD resulting from the conversion. It indicates the magnitude of error introduced by such a conversion. The sample names of the mixtures reflect the order of magnitude of volumetric peat content.

The WRC of all mixtures were determined by using pressure plate extractors at seven different matric potentials: -1, -10,

-31.6, -100, -158.5, -1000 and $-15\,848.9\,\text{cm}$. Willaredt and Nehls (2021) used different binary mixtures of ground bricks (GB) and green waste compost (GWC) with volumetric shares of GWC of 0, 18, 28, 37, 47, 68, 100 % (volume/volume). The





**Table 1.** Converted volumetric share of peat derived from mass specific mixing ratio and magnitude of resulting error

| Sample | $x_{i,v}$ [cm$^3$ cm$^{-3}$] | $x_{i,m}$ [g g$^{-1}$] | BD$_{meas}$ [g cm$^{-3}$] | BD$_{calc}$ [g cm$^{-3}$] |
|---|---|---|---|---|
| P0S10 | 0 | 0 | 1.86 | 1.86 |
| P2S8 | 0.24 | 0.5 | 1.57 | 1.49 |
| P6S4 | 0.64 | 0.2 | 1.05 | 0.88 |
| P8S2 | 0.82 | 0.4 | 0.68 | 0.61 |
| P9S1 | 0.93 | 0.6 | 0.51 | 0.44 |
| P99S01 | 0.99 | 0.8 | 0.41 | 0.35 |
| P10S0 | 1 | 1 | 0.33 | 0.33 |

**Table 2.** Properties of components constituting the investigated binary mixtures. Porosity, if not provided, was calculated from particle density, bulk density and soil sample volume.

| Property | | Willaredt & Nehls 2021 | | Walczak et al. 2002 | | Deeb et al. 2016 | | Sakaki & Smits 2015 | |
|---|---|---|---|---|---|---|---|---|---|
| | | GB | GWC | S | P | EDH | GWC | CS | FS |
| BD | [g cm$^{-3}$] | 1.35 | 0.64 | 1.86 | 0.33 | 1.17 | 0.37 | 1.77 | 1.74 |
| PD | [g cm$^{-3}$] | 2.63 | 2.32 | NA | NA | 2.75 | 2.06 | 2.65 | 2.65 |
| C concentration | [g kg$^{-1}$] | 24 | 268 | 1 | 574 | 0.38 | 214 | NA | NA |
| porosity | [m$^3$ m$^{-3}$] | 0.49 | 0.69 | 0.38 | 0.9 | 0.57 | 0.82 | 0.34 | 0.34 |

GB: ground bricks, GWC: green waste compost, P: peat, S: sand, EDH: excavated deep soil horizon, CS: coase sand, FS: fine sand

respective denomination refers to the rounded bulk volumetric share of GWC: C0B10, C2B8, C3B7, C4B6, C5B5, C7B3 and C10B0 The water retention curves of 5 replicates of each mixture was measured combining the simplified evaporation method (Schindler, 1980; Peters et al., 2015), using the HYPROP© device (Metergroup, Munich, Germany) and the dew point method
(Campbell et al., 2007) using the WP4C device (Metergroup, Munich, Germany). For details of the measurements and the data evaluation, the reader is referred to Willaredt and Nehls (2021). Some basic properties are summarized in Table 2. Sakaki and Smits (2015) combined coarse sand (mean grain size D = 1.04 mm) and fine sand (mean grain size d = 0.12 mm), thus choosing two components with a pronounced difference in particle size. They obtained water retention measurements of a high resolution for matric potentials ranging between 1 and 135 cm using an induced drainage process in a modified Tempe cell
setup Sakaki and Illangasekare (2007). Table 2 summarizes selected properties of the components used for composing each of the four data sets.





## 2.4 Fitted water retention models

We used the SHIPFIT2.0 software implemented in HYPROP-Fit (Pertassek et al., 2015) to fit parametric water retention models to the data. For each data set we chose the model presenting the best performance in regard to matching the observations in the respective measurement range without over parameterization. The data of Willaredt and Nehls (2021) was measured in high resolution and showed a complex pore structure, the PDI (Peters, 2013; Iden and Durner, 2014; Peters, 2014) model with the unconstrained bimodal van Genuchten basic function (Van Genuchten, 1980) was fitted to the data. The model was fitted to all replicates of each mixture. Due to its limited matric potential range but yet high resolution (Fig. 4), the data sets of Sakaki and Smits (2015) were described with the PDI model using the constrained bimodal van Genuchten function (Durner, 1994). The data sets of Deeb et al. (2016) and Walczak et al. (2002) have less observations (n=9 and n=7, respectively for each subset), therefore unimodal models were applied. The data set by Deeb et al. (2016) was best represented by the PDI model with the unimodal constrained van Genuchten function (Van Genuchten, 1980) as basic function, whereas the data set of Walczak et al. (2002) was best represented using the original unimodal constrained model of Van Genuchten (1980). The latter can be explained by he comparably high remaining water contents at high matric potentials. The detailed model descriptions and the obtained parameters together with the RMSE between the models and observations are summarized in the Appendix (Table A1-A4).

## 2.5 Evaluation of predictions

We evaluate the predictive performance of the described compositional model approaches by calculating the RMSE between the fitted and the predicted curves:

$$\text{RMSE} = \sqrt{\frac{1}{r} \sum_{i=1}^{r} \left( \theta_{\text{fit}} - \theta_{\text{pred}} \right)^2}. \tag{5}$$

Where $\theta_{\text{fit}}$ [-] is the water content at the specific matric potential given by the fitted curve, $\theta_{\text{pred}}$ [-] is the predicted water content using one of the compositional models and r is the number points on the curves used. We furthermore analyze the absolute deviation between the fitting models and predictions for every matric potential. It is calculated as the difference between the modeled and fitted water contents at similar matric potentials, meaning that positive deviations indicate that the prediction overestimates the water contents compared to value of the fitted curve and negative values vice versa.

## 2.6 Estimation of distribution of water and air in constructed Technosols

Based on the predicted and fitted water retention curves we calculate the distribution of air and water content in hydrostatic equilibrium assuming the application case of the substrate in a small-scale green infrastructure element having an established soil depth of $0.5\,\text{m}$. Thus, the matric potential, considered as container capacity, corresponds to a value of approximately pF 1.7. The air content is simply given by $\phi - \theta$ , where $\phi$ is the porosity, which was either provided in the original articles or calculated from the respective mean bulk density and particle density.





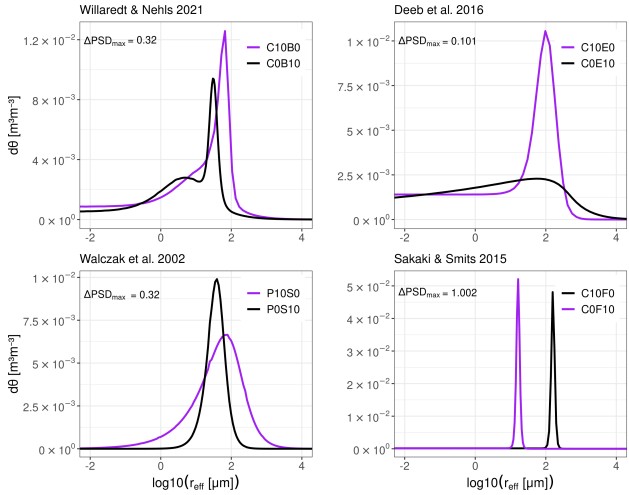

**Figure 3.** Pore size distribution of each component used to compose the investigated mixtures. The magnitude of the distance between each curves maxima $\Delta\mathrm{PSD_{max}}$ describes the size difference of the most abundantly occurring pores in both components.

## 3 Results & Discussion

### 3.1 Pore size distribution in components of binary mixtures

The pore size distribution of the single components for Technosol construction provides a useful measure to chose the right type

of model to predict the Technosol's water retention curves. Figure 3 assembles the PSD curves computed for all components combined to binary mixtures. Each plot is supplemented with value of $\Delta\mathrm{PSD_{max}}$. That quantifies the order of magnitude laying between the size of the most abundantly occurring pore sizes in both components.

In the data of Willaredt and Nehls (2021) the pore size corresponding to the maxima of the PSD in green waste compost (C10B0) is approximately twice as big as the $\mathrm{PSD_{max}}$ in ground bricks (C0B10). The sand (P0S10) and peat (P10S0) chosen

for the mixtures prepared by Walczak et al. (2002) show a similar difference. The smallest difference was determined for the excavated deep soil (C0E10) and green waste compost (C10E0) (Deeb et al., 2016) with the most abundantly present pores in green waste compost only 1.26 times larger than those on the excavated deep soil horizon. The most pronounced difference between the PSDmax was determined for the mixture of coarse sand (C10F0) and fine sand (C0F10) investigated by Sakaki and Smits (2015). Here the size difference between the most abundantly occurring pore size in coarse sand is 10 times

larger than the dominant pore size found in fine sand. The PSD of the components that are relevant for Technsol construction (GWC, ground bricks, peat, sand and excavated deep soil horizon material) show small differences between $\mathrm{PSD_{max}}$. Hence, the difference between them is too small and the two systems will not interlock as it would be the case for the fine and coarse sand by Sakaki and Smits (2015) (compare Fig. 1). Based on these differences the model type can be selected. The predictions for the data sets by Willaredt and Nehls (2021), Walczak et al. (2002) and Deeb et al. (2016) were predicted using the „zero





mixing" approach, hence the basic compositional models. The model type „ideal mixing" was applied to the data by Sakaki and Smits (2015).

## 3.2  Impact of data quality and resolution

The pore size distributions in Fig. 3 show bi-modality only for the data set of Willaredt and Nehls (2021), most probably due to the high resolution of the water retention curve. Therefore, a bimodal parametric model was chosen to represent the

water retention curve. The bi-modality is more severe for the ground bricks (C0B10), most likely due to their internal porosity, that was found present for ground brick particles bigger than $0.2\,\text{mm}$ (Nehls et al., 2013). However, the green waste compost (C10B0) also reveals a secondary pore system with most pores having the size of approximately $1\,\mu\text{m}$. It is likely that the green waste compost used in the mixtures formulated by Deeb et al. (2016) presents a similar structure, however due to the comparably small number of observations on the curve, such a structure remains undiscovered. We therefore stress the

importance of high-resolution measurements and a wide range of matric potentials on which the presented predictions of water retention curves of the mixtures should be based on. The evaporation method implemented in the HYPROP© device accounts for high resolution measurements, however the measurement range here should be extended towards higher matric potentials by complementary measurements, e.g. with the WP4C dewpoint water potential meter (Flores-Ramírez et al., 2018). Furthermore, we would like to outline the need for a systematic measurement campaign of water retention curves of materials, found to be

suitable components in Technosol construction (e.g. Rokia et al. (2014). A comprehensive database would be helpful for further validation of the described concept regarding similarity in PSDs. So far, this similarity is a qualitative description and a more precise quantification of $\Delta\text{PSD}_{\text{max}}$ should be addressed based on a comprehensive data base.

## 3.3  Predicted water retention curves

The plots in Fig. 4-7 illustrate the water retention curves described by the fitted parametric model next to the predicted water

retention curves. The first panel in each plot shows the curves of the pure components, used as model input. The curves are presented together with the corresponding RMSE, that quantifies the average deviation between the predictions and the fitted curves.

The adapted Clarke model, is well-suited to predict the data by Sakaki and Smits (2015) in fine-controlled mixtures. The mixtures C2F8, C5F5 and C7F3 are considered fine controlled. For coarse-controlled mixtures the Clarke model accounts well

for the observations in the wet range, which is not surprising as it was adapted from the model for porosity prediction. Whereas the air entry point in the mixture C9F1 is not impacted by the small volumetric share of fine sand, it does introduce a difference for C8F2. Neither the Clarke approach nor the basic or extended compositional model predict the impact of the addition of small amounts of fine sand to the mixture properly. This can be explained by the heterogeneity in such a mixture that develops when a part of the pores formed by the large particles is filled with fine particles and another part remains empty (Naseri et al.,

2019). Mixtures of coarse and fine sand are not relevant for Technosol constructions in practice. However popular commercial constitutes in green roof media and horticultural substrates are coarse technically expanded geogenic particles presenting intra-porosity (Hill et al., 2019). The description of their water retention characteristics by Flores-Ramírez et al. (2018) show a clear



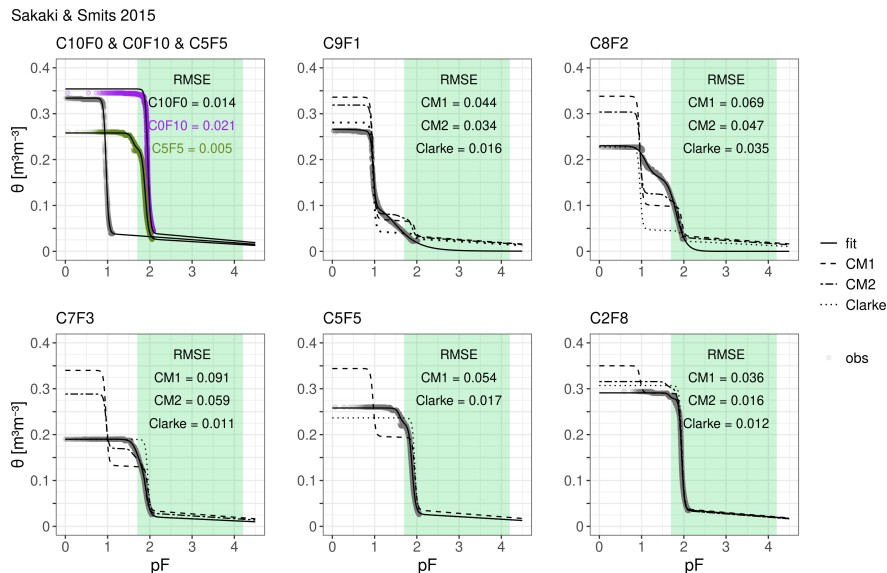

**Figure 4.** Water retention curves of all 7 binary mixtures produced from coarse sand (C10F0) and fine sand (C0F10). Observations are represented by gray dots, the fitted parametric representations are represented by the solid line and the predicted curves are represented by the dashed lines. CM1 stands for the basic compositional model and CM2 for the extended scheme, Clarke stands for the adapted model from Clarke (1979). The first panel (top, left) assembles the water retention curves of the pure components and the intermediate mixture, which constitute the input for the prediction model. The particular RMSE describes the deviation between the predictions and the fitted curves. Note that C5F5 is not predicted by the extended model scheme because it is considered the intermediate mixture.

bimodal pore structure. For constructed Technosols containing such, the Clarke model could be applied in a modified version that accounts for water retention within the coarse particles.

For the data set of Willaredt and Nehls (2021) the fitted parametric model curves (Fig. 5) are characterized by RMSEs ranging between $0.005\,\mathrm{m^3\,m^{-3}}$ for the mixture C4B6 in the best case and $0.02\,\mathrm{m^3\,m^{-3}}$ for the mixture C5B5 in the worst case. The averaged deviation between the predicted WRC and fitted WRC is generally smaller than $2\,\%$. Using the extended scheme improves the prediction regarding the RMSE in three of four cases (mixture C4B6, C3B7 and C2B8). We find similar well representations of the data observed by Walczak et al. (2002) (Fig. 6). Here the predictions show RMSE between the fitted

and predicted curves ranging from $0.01\,\mathrm{m^3\,m^{-3}}$ to $0.03\,\mathrm{m^3\,m^{-3}}$ having the same order of magnitude as the errors calculated between the observations and corresponding parametric representations, ranging from $0.006\,\mathrm{m^3\,m^{-3}}$ to $0.029\,\mathrm{m^3\,m^{-3}}$ (Table A4). Using the extended scheme for this data set improves the representation in the average for the mixtures P2S8, P8S2, P9S1. The improvements with CM2 are especially observable in the dry end of the WRC (pF > 1.2). The deviations here reflect the





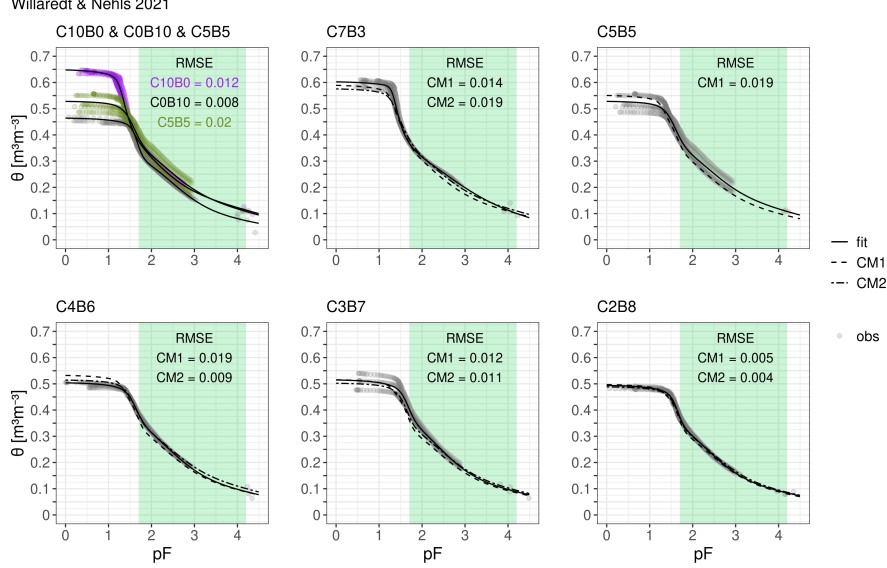

**Figure 5.** Water retention curves of all 7 binary mixtures of ground bricks (C0B10) and green waste compost (C10B0). Observations are represented by gray dots, the fitted parametric representations are represented by the solid line and the predicted curves are represented by the dashed lines. CM1 stands for the basic compositional model and CM2 for the extended scheme. The first panel (top, left) assembles the water retention curves of the pure components and the intermediate mixture, which constitute the input for the prediction model. The particular RMSE describes the deviation between the predictions and the fitted curves. Note that C5B5 is not predicted by the extended model scheme because it is considered the intermediate mixture.

fitting quality of the parametric model used to represent the data of the pure peat (RMSE $0.029\,\mathrm{m^3\,m^{-3}}$) for the pure peat. This

leads to deviations in the predictions that tend to be corrected if the extended scheme is applied.

### 3.4 Absolute deviations along the water retention curve

The RMSE as a measure, that averages occurring deviations for all matric potentials can mask the bad performance of the predictions in some parts of the curve. Therefore, the consideration of the absolute deviations (compare Fig. 8) over different matric potentials complements the assessment. Generally for the data set of Willaredt and Nehls (2021), over all matric po-

tentials the deviation is largest in the wet range and does not exceed 4.2 %. In the wet range the predictions made using the basic compositional model approach (CM1) tend to overestimate the water contents. Compared to that the extended approach underestimates the water contents in the same range and diminishes the absolute deviation here only for the mixture C4B6, which presents the closest mixing ratio to the intermediate mixture. For the mixtures in data set of Walczak et al. (2002) that contain volumetric shares of peat $x_{i,v} > 0.6$ the extended scheme CM2 leads to smaller deviations in the dry range.





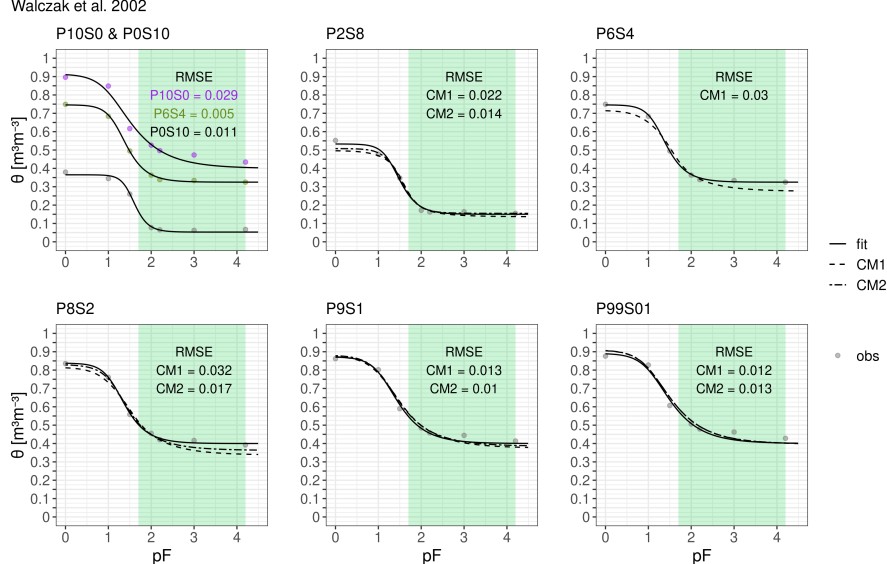

**Figure 6.** Retention curves of all 7 binary mixtures of sand (P0S10) and peat (P10S0). Observations are represented by gray dots, the fitted parametric representations are represented by the solid line and the predicted curves are represented by the dashed lines. CM1 stands for the basic compositional model and CM2 for the extended scheme. The first panel (top, left) assembles the water retention curves of the pure components and the intermediate mixture, which constitute the input for the prediction model. The particular RMSE describes the deviation between the predictions and the fitted curves. Note that P6S4 is not predicted by the extended model scheme because it is considered the intermediate mixture.

Obviously the method used for determining the water retention curves of the main components has an impact on the prediction quality. The case of a larger deviation of the observed water contents in replicates leads to poor representations by the parametric fits that are used to predict water retention curves of other mixtures. On one hand the deviation between the replicates of the components introduces an error when being used as model input for predicting the WRC of the mixtures. On the other hand, the deviation resulting from the uncertainties of sample preparation of any mixture also defines the magnitude
of the tolerable error when predicting the curves by the means of our model approach. The tested data sets of Deeb et al. (2016) and Willaredt and Nehls (2021) were derived from replicated observations (compare Fig. 7 and 5). In addition to the RMSEs summarized in the corresponding Fig.s, Table 3 provides the absolute maximal deviations, and minimal deviations respectively for each observed mixture from the parametric representation in the data set of Deeb et al. (2016) and Willaredt and Nehls (2021), thus providing the magnitude of the tolerable error by our predictions.

The values are bigger for the data set obtained by Deeb et al. (2016) using a more practice-oriented sampling strategy from containers. Along the observed pressure head range the biggest deviations occur in the mixture C5E5. Here the parametric fit



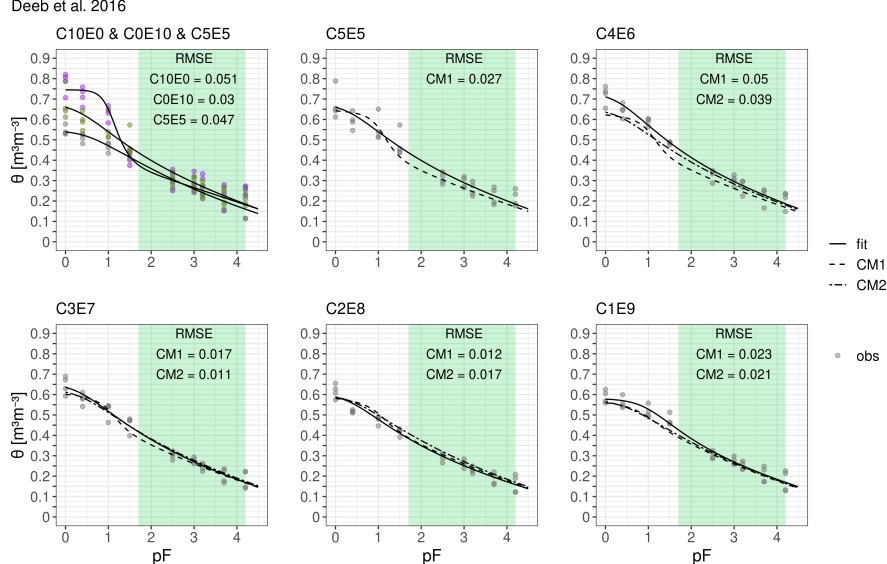

**Figure 7.** Water retention curves of all 7 binary mixtures of excavated deep soil horizon (C0E10) and green waste compost (C10E0). Observations are represented by gray dots, the fitted parametric representations are represented by the solid line and the predicted curves are represented by the dashed lines. CM1 stands for the basic compositional model and CM2 for the extended scheme. The first panel (top, left) assembles the water retention curves of the pure components and the intermediate mixture, which constitute the input for the prediction model. The particular RMSE describes the deviation between the predictions and the fitted curves. Note that C5E5 is not predicted by the extended model scheme because it is considered the intermediate mixture.

underestimated the observed water contents in the worst case by 13 %. The deviations remain similarly large along all observed matric potentials. Following the sampling preparation protocol introduced by Willaredt and Nehls (2021) yields comparably smaller deviations related to differing bulk densities. Here the biggest misfit was observed for the mixture C7B3, were the

parametric representation underestimates the observation by 5 %. In Fig. 8 it can be observed that the deviations decrease for higher tensions, except for the mixture C5B5. According to (Jackisch et al., 2020), this reflects a deviation related to different bulk densities of samples that are homogeneous otherwise. However, the deviations related to different compaction of Technosols when used in practice are expected to be larger. Figure 8 visualizes the absolute deviations between the predictions and the parametric representations for all predicted WRC. In the data sets from Willaredt and Nehls (2021) as well as from

Deeb et al. (2016) the deviations remain smaller than the maximum deviations described in the section above.



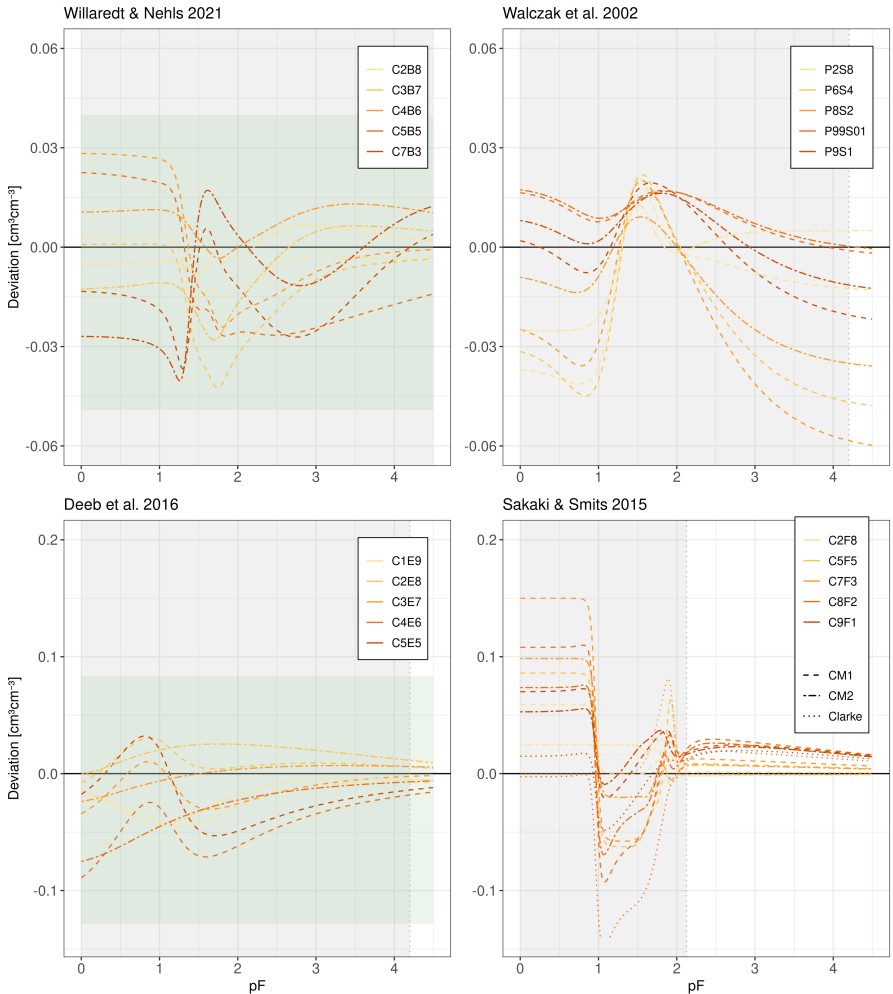

**Figure 8.** Absolute deviation between predicted and observed and fitted water contents over different matric potentials. The shaded pressure head range in gray was covered by measurements. The solid line represents the deviation between predictions with the basic model scheme (CM1) and the parametric fittings. The dashed line represents the deviation between the extended model scheme (CM2) and the parametric fittings. The green ribbon illustrates the maximum deviation that occurred between the observations and fitted representations.

## 3.5 Comparison of basic and extended scheme

The plots in Fig. 8 show that the largest deviations result in the wet range, except for the data set of Walczak et al. (2002). The extended model approach leads not only to smaller RMSE but also to smaller absolute deviations. Some exceptions exist. Nevertheless, the curves predicted using the basic compositional model approach are already representing the observations
in a quality that does not justify further improvement by more laboratory work. However, an additional measurement of an





**Table 3.** Maximum and minimum deviation between observations and fitted parametric representation of volumetric water contents of all observed matric potentials. The magnitude reflects the differences between the replicates because of different sampling strategies (packing cylinders to a defined weight for compaction vs. in situ sampling from containers).

| | Willaredt & Nehls 2021 | | | Deeb et al. 2016 | |
| Mixture | Min deviation $[\mathrm{m^3\,m^{-3}}]$ | Max deviation $[\mathrm{m^3\,m^{-3}}]$ | Mixture | Min deviation $[\mathrm{m^3\,m^{-3}}]$ | Max deviation $[\mathrm{m^3\,m^{-3}}]$ |
|---|---|---|---|---|---|
| C0B10 | -0.04 | 0.04 | C0E10 | -0.06 | 0.05 |
| C2B8 | -0.01 | 0.01 | C1E9 | -0.06 | 0.04 |
| C3B7 | -0.03 | 0.04 | C2E8 | -0.07 | 0.05 |
| C4B6 | -0.02 | 0.02 | C3E7 | -0.05 | 0.06 |
| C5B5 | -0.04 | 0.04 | C4E6 | -0.05 | 0.07 |
| C7B3 | -0.05 | 0.02 | C5E5 | -0.13 | 0.08 |
| C10B0 | -0.04 | 0.02 | C10E0 | -0.09 | 0.08 |

intermediate mixture can always serve as a validation measurement, proving that the approach does not fail for the components chosen for the Technosol formulation.

### 3.6 Selecting Technosol recipes based on predicted WRCs

Based on the predicted water retention curves it is possible to analyze and compare the performance of Technosols e.g. as planting substrates formulated as binary mixtures 1) in any possible mixing ratio and 2) from different components. The first type of comparison provides the ability to narrow down the infinite options provided by combining two components to a full range of mixtures to such mixing ratios, that perform as desired. The second type of comparison provides the ability to select the most suitable component from those available and to exclude components that do not feature plant growth supporting properties. As an exemplary case, we calculated the distribution of water and air contents in two of the investigated binary mixtures, analyzed in this study. In the case we assume that the Technosols are implemented as planting substrates in a container of 0.5 m depth. We chose those two binary mixtures, that are formulated of alternative materials likely to be used as constitutes for Technosols in urban green infrastructure applications: the mixture of excavated deep soil horizon together with green waste compost (Deeb et al., 2016) and the mixture of ground bricks and green waste compost (Willaredt and Nehls, 2021). The distribution of air throughout the depth of the planting container determines the favorable rooting depth in a container filled with the Technosols. From the results illustrated in Fig.9, we conclude that green waste compost introduces the pore space to the mixture that is needed to guarantee supply of air for the roots in shallow containers. For Technosols that include ground bricks as a mineral component, the GWC content should be at least 50 vol% to omit insufficient air supply in the root zone. Technosols formulated with excavated deep soil present sufficient supply of air in containers when containing 20 vol% GWC, this confirms the results by (Deeb et al., 2016). Caron et al. (2015) assess the effect of evolution in growing substrate

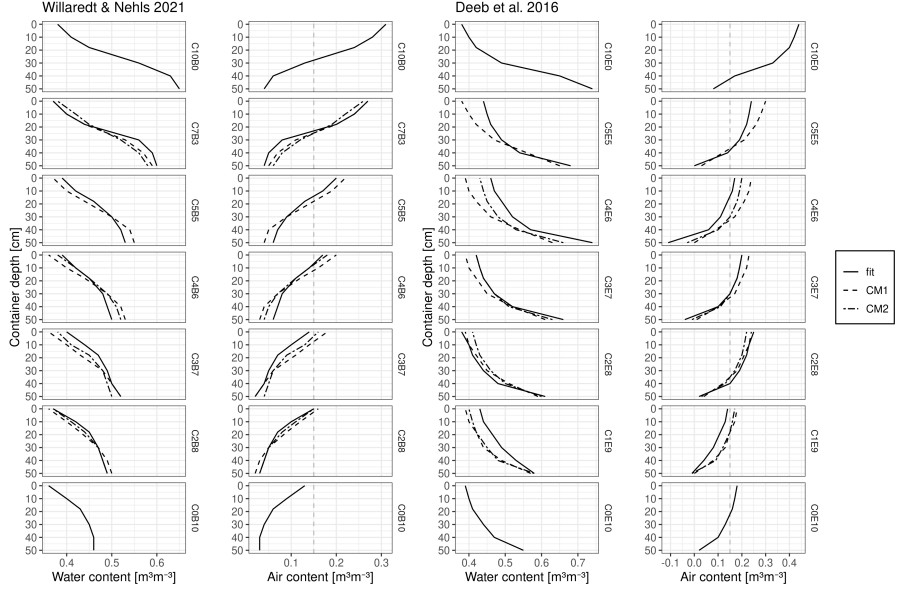

**Figure 9.** Distribution of volumetric water and air content over different depths at hydro static equilibrium in a container (pF = 1.7) filled with a constructed Technosol formulated as a binary mixture of: green waste compost and ground bricks (left) and green waste compost and excavated deep soil (right). The solid line indicates the fitted WRCs, the dashed lines indicate predictions using the basic scheme (CM1) or the extended scheme (CM2) respectively. The grey vertical line indicates the minimum volumetric air content in horticultural substrates favorable for root growth (Caron et al., 2015).

due to differences in compaction when filling the containers, disturbances and root development on the substrate properties in containers as so severe, that they recommend the assessment of properties within the container. They especially stress the availability of oxygen in substrates containing a high share of organic material as a limiting factor.

For some mixtures by Deeb et al. (2016) we yield negative air contents, since we calculated the contents based on mean bulk densities. The description of the WRC is required to facilitate numerical simulations for analyzing the hydrological 310 behavior of a constructed Technosol e.g. under real climatic conditions (Brunetti et al., 2016). For those models that require parametric representations, a post-hoc fitting will be required. Simulation based choices for Technosol compositions would lift the application of constructed Technsols in urban green infrastructure onto a new level.

## 4 Conclusions

This study presents a compositional model that allows to predict the water retention curve (WRC) of a constructed Technosol 315 formulated as a binary mixture for any mixing ratio. The predictions are based on the measured WRC of the pure components and the volumetric mixing ratio. Thus, only a small measurement effort is required for describing a large number of possible combinations. The model was shown to be applicable to mixtures of components that are characterized by a small difference





between their pore space distribution maxima ($\Delta\text{PSD}_{\text{max}}$). It can be concluded that the model performs best based on water retention observations that have a high reproducibility, a high resolution and that cover a large range of pressure heads. The latter is essential when it is aimed to use the predicted curves for numerical simulations. From the comparison between predicted and fitted WRCs of three case study mixtures that are of practical relevance for Technosol construction we conclude, that the approach should be valid for further materials and their compositions. As a practical application of the predicted WRCs the hydrostatic distribution of water and air in constructed Technsols was demonstrated. That facilitates optimizing the choice of the components and its mixing ratios or the constructed soil depth. The results of this study indicate the added value of a systematic soil physical characterization of potential Technosol components e.g. in form a database. Such data could be used to further evaluate the presented approach and for theoretical experiments searching for purpose-designed Technosol recipes. The proposed model is a milestone on the path towards simulation-based design of Technosols providing specific soil functions.

*Data availability.* In the appendix we provide the fitting parameters and WRC models used to represent the water retention data sets presented in this study. The raw data from third parties can not be made available. The raw data related to the work by Willaredt and Nehls (2021) can be obtained upon request from the corresponding author.

## Appendix A: Description of fitted water retention models

The data of Willaredt and Nehls (2021) was represented with the PDI (Peters, 2013; Iden and Durner, 2014; Peters, 2014) model with the unconstrained bimodal (Durner, 1994) basic function of Van Genuchten (1980), the respective parameters are displayed in Table A1. The PDI model accounts for both capillary and adsorptive water retention ($S^{cap}$ [-] and $S^{ad}$ [-]) :

$$\theta\left(h\right) = \left(\theta_s - \theta_r\right) \cdot S^{cap} + \theta_r S^{ad}. \tag{A1}$$

$\theta\left(h\right)$ stands for the volumetric water content [$\text{m}^3\,\text{m}^{-3}$] and h [cm] stands for the matric potential. To ensure that water content is 0 for $h = h_0 = 10^{6.8}$ the basic function in the capillary saturation function is scaled as follows:

$$S^{cap}\left(h\right) = \frac{\Gamma\left(h\right) - \Gamma_0}{1 - \Gamma_0} \tag{A2}$$

The constrained retention function of van Genuchten (1980) is described by

$$\Gamma\left(h\right) = \left[\frac{1}{1 + \left(\alpha h\right)^n}\right]^{1 - \frac{1}{n}}. \tag{A3}$$

$\alpha$ [$\text{cm}^{-1}$] and n [-] are curve shape parameters. The unconstrained function of van Genuchten (1980) is described by:

$$\Gamma\left(h\right) = \left[\frac{1}{1 + \left(\alpha h\right)^n}\right]^{m}. \tag{A4}$$





where m [-] stands for an additional shape parameter. In the bimodal form of (Durner, 1994) the basic functions are weighted and added:

$$\Gamma(h) = \sum_{i=1}^{2} w_i \Gamma_i \tag{A5}$$

with $w_i$ standing for the weighting factor of the sub functions, with $0 < w_i < 1$ and $\sum w_i = 1$. The adsorptive water retention is calculated as:

$$S^{ad}(x) = 1 + \frac{1}{x_a - x_0} \left( x - x_a + b \, ln \left[ 1 + \exp\left( \frac{x_a - x}{b} \right) \right] \right) \tag{A6}$$

where the smoothing parameter b for the adsorption function in the constrained van Genuchten function is calculated with:

$$b = 0.1 + \frac{0.2}{n^2} \left[ 1 - \exp\left( -\frac{\theta_r}{\theta_s - \theta_r} \right) \right]^2 \tag{A7}$$

and for the unconstrained van Genuchten function with:

$$b = 0.1 + 0.07\sigma \left[ 1 - \exp\left( -\frac{\theta_r}{\theta_s - \theta_r} \right) \right]^2 \tag{A8}$$

## A1 Fitting parameters

In the following the fitting parameters obtained for every mixture of each data set are presented in Tables A1-A4 with the
corresponding RMSE as a diagnostic variable describing the mean deviation between the fitted model and the observation. The data sets of Sakaki and Smits (2015) was described with the PDI model using the constrained bimodal van Genuchten function (Durner, 1994). The respective parameters are displayed in Table A2.

The data set of Deeb et al. (2016) was represented using the PDI model with the unimodal constrained van Genuchten function as basic function, the respective parameters are displayed in Table A3.

The data set of Walczak et al. (2002) was represented using the original unimodal constrained model of van Van Genuchten (1980), the respective parameters are displayed in Table A4.

*Author contributions.* Conceptualization, M.W. and T.N.; Model implementation and Analysis, M.W., A.P. and T.N. writing - original draft preparation, M.W; writing - review and editing, M.W., T.N., A.P. ; All authors read and approved the final manuscript.

*Competing interests.* The authors declare that there are no competing interests.



**Table A1.** Fitted Parameters to water retention observations from Willaredt & Nehls 2021, bimodal PDI unconstrained van Genuchten variant and RMSE between model and observations.

| Mixture | $x_{i,v}$ [$m^3\,m^{-3}$] | $\alpha_1$ [$cm^{-1}$] | $n_1$ [-] | $\theta_r$ [$m^3\,m^{-3}$] | $\theta_s$ [$m^3\,m^{-3}$] | $\alpha_2$ [$cm^{-1}$] | $n_2$ [-] | $w_2$ [-] | $m_1$ [-] | $m_2$ [-] | RMSE [$m^3\,m^{-3}$] |
|---|---|---|---|---|---|---|---|---|---|---|---|
| C0B10 | 0 | 0.00335 | 0.933 | 0.134 | 0.465 | 0.0213 | 5.902 | 0.361 | 1 | 1 | 0.008 |
| C2B8 | 0.18 | 0.00448 | 0.963 | 0.159 | 0.495 | 0.0224 | 5.204 | 0.342 | 1 | 1 | 0.006 |
| C3B7 | 0.28 | 0.00442 | 0.952 | 0.166 | 0.516 | 0.0211 | 4.462 | 0.314 | 0.999 | 1 | 0.015 |
| C4B6 | 0.37 | 0.00404 | 0.932 | 0.168 | 0.505 | 0.0231 | 3.856 | 0.347 | 1 | 1 | 0.005 |
| C5B5 | 0.47 | 0.00413 | 0.926 | 0.209 | 0.529 | 0.0257 | 3.468 | 0.433 | 1 | 1 | 0.02 |
| C7B3 | 0.68 | 0.0054 | 0.824 | 0.148 | 0.604 | 0.0473 | 9.382 | 0.495 | 0.531 | 0.228 | 0.014 |
| C10B0 | 1 | 0.00935 | 0.968 | 0.237 | 0.65 | 0.0514 | 6.879 | 0.515 | 1 | 0.346 | 0.012 |

**Table A2.** Fitted Parameters to water retention observations from Sakaki and Smits (2015), bimodal PDI constrained van Genuchten variant

| Mixture | $x_{i,v}$ [$m^3\,m^{-3}$] | $\alpha_1$ [$cm^{-1}$] | $n_1$ [-] | $\theta_r$ [$m^3\,m^{-3}$] | $\theta_s$ [$m^3\,m^{-3}$] | $\alpha_2$ [$cm^{-1}$] | $n_2$ [-] | $w_2$ [-] | RMSE [$m^3\,m^{-3}$] |
|---|---|---|---|---|---|---|---|---|---|
| C0F10 | 0 | 0.0112 | 15 | 0.04 | 0.354 | 0.00143 | 8.701 | 0 | 0.021 |
| C2F8 | 0.2 | 0.0113 | 15 | 0.039 | 0.291 | 0.0221 | 14.057 | 0.046 | 0.009 |
| C5F5 | 0.7 | 0.013 | 11.104 | 0.029 | 0.258 | 0.0258 | 10.319 | 0.148 | 0.005 |
| C7F3 | 0.7 | 0.0123 | 9.857 | 0.022 | 0.19 | 0.0199 | 5.937 | 0.429 | 0.002 |
| C8F2 | 0.8 | 0.0852 | 4.698 | 0 | 0.23 | 0.0162 | 4.341 | 0.699 | 0.004 |
| C9F1 | 0.9 | 0.1089 | 15 | 0.001 | 0.266 | 0.043 | 2.171 | 0.388 | 0.005 |
| C10F0 | 1 | 0.1092 | 15 | 0.039 | 0.334 | 0.00049 | 1.02 | 0 | 0.014 |

*Acknowledgements.* We would like to thank Maha Deeb, Toshihiro Sakaki and Kathleen M. Smits for providing water retention data. M. Willaredt thanks the Berlin International Graduate School on Model and Simulation Based Research (BIMoS) for funding her dissertation project. T. Nehls thanks the BMWi (Ugreen, FKZ 03EN1045C). A.Peters thanks the DFG (Deutsche Forschungsgemeinschaft (DFG grant PE 1912/4-1)). We thank Sarah Sanford for linguistic revision.





**Table A3.** Fitted Parameters to water retention observations from Deeb et al. (2016), unimodal PDI constrained van Genuchten variant

| Mixture | $x_{i,v}$ [$\mathrm{m^3\,m^{-3}}$] | $\alpha$ [$\mathrm{cm^{-1}}$] | n [-] | $\theta_r$ [$\mathrm{m^3\,m^{-3}}$] | $\theta_s$ [$\mathrm{m^3\,m^{-3}}$] | RMSE [$\mathrm{m^3\,m^{-3}}$] |
|---|---|---|---|---|---|---|
| C0E10 | 0 | 0.31 | 1.109 | 0.161 | 0.551 | 0.03 |
| C1E9 | 0.1 | 0.1144 | 1.336 | 0.345 | 0.581 | 0.029 |
| C2E8 | 0.2 | 0.5 | 1.235 | 0.33 | 0.609 | 0.028 |
| C3E7 | 0.3 | 0.5 | 1.208 | 0.326 | 0.658 | 0.031 |
| C4E6 | 0.4 | 0.5 | 1.258 | 0.4 | 0.737 | 0.032 |
| C5E5 | 0.5 | 0.5 | 1.245 | 0.4 | 0.682 | 0.047 |
| C10E0 | 1 | 0.0843 | 2.949 | 0.4 | 0.745 | 0.05 |

**Table A4.** Fitted Parameters to water retention observations from Walczak et al. (2002), original unimodal constrained van Genuchten model

| Mixture | $x_{i,v}$ [$\mathrm{m^3\,m^{-3}}$] | $\alpha$ [$\mathrm{cm^{-1}}$] | n [-] | $\theta_r$ [$\mathrm{m^3\,m^{-3}}$] | $\theta_s$ [$\mathrm{m^3\,m^{-3}}$] | RMSE [$\mathrm{m^3\,m^{-3}}$] |
|---|---|---|---|---|---|---|
| P0S10 | 0 | 0.0295 | 3.148 | 0.053 | 0.365 | 0.011 |
| P2S8 | 0.24 | 0.0447 | 2.482 | 0.15 | 0.533 | 0.017 |
| P6S4 | 0.64 | 0.058 | 2.307 | 0.325 | 0.746 | 0.006 |
| P8S2 | 0.82 | 0.0682 | 2.144 | 0.4 | 0.838 | 0.008 |
| P9S1 | 0.93 | 0.071 | 1.74 | 0.4 | 0.872 | 0.017 |
| P99S01 | 0.99 | 0.0753 | 1.881 | 0.4 | 0.891 | 0.025 |
| P10S0 | 1 | 0.0839 | 1.641 | 0.4 | 0.914 | 0.029 |

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
