# Peer review of "Predicting Soil Hydraulic Properties for Binary Mixtures - Concept and Application for Constructed Technosols"

_Hydrology and Earth System Sciences, 2022_

## Referee Comment (RC2)

[referee-annotated manuscript omitted]

---

## Author Comment (AC1)

*Reply to anonymous Reviewer #1*

**RC1 #1: The motivation and objectives of this work are clearly described.**

We thank for the dedicated work and time to review our manuscript and appreciate the positive feedback regarding our motivation and objectives.

**RC1 #2: Compared to the usual high scientific level of papers published in HESS, the paper should be improved by addressing also the saturated hydraulic conductivity.**

We agree with the reviewer that considering the hydraulic conductivity (not only saturated) will substantially improve the manuscript and highlight the indented hydrological focus of our study. We are aware of the particular importance of soil hydraulic properties for numerical simulations of flow and transport processes in the soil-plant-atmosphere system. In the revised manuscript we will use the theory developed by Peters et al. (2023) (HESSD https://hess.copernicus.org/preprints/hess-2022-431/), which does not require any conductivity measurements because these data is missing for most of the used data sets.

**RC1 #3: It should also provide a deeper discussion of the model parameter xcrit whose impact on the model results remains unclear to me.**

We agree with the reviewer, that the model parameter $x_{crit}$ is not discussed deeply and are thankful for that comment. It reveals the potential to improve the formulation of our adapted Clarke model by reviewing the parameter $x_{crit}$. The parameter $x_{crit}$ originates from the work on porosity in binary mixtures of coarse and fine particles of Clarke (1979) and refers to a critical volumetric fraction of fine particles. According to his concept (confirmed by later studies e.g. Zhang et al. (2011) and Sakaki and Smits (2015)), $x_{crit}$ corresponds to the porosity of the coarse fraction. Therefore, introducing the extra parameter $x_{crit}$ is not required and we suggest simplifying the model by replacing it now with $\varphi_c$, which is the porosity of the coarse fraction. This critical content can be understood as a threshold at which the relation between the porosity and the volumetric fraction of the fine particles changes from "coarse-controlled" to "fine-controlled" (Sakaki and Smits, 2016). These two regimes are equally observable in the water retention data of binary mixtures of coarse and fine particles (Sakaki and Smits, 2016). We accordingly implemented these two regimes in our adapted Clarke model by providing two equations for each respective case.
In the revised manuscript we suggest expressing the parameter $x_{crit}$ explicitly by replacing it with the porosity of the coarse fraction $\varphi_c$ leading to:

$$\theta_{pred}=\begin{cases} \left(x_f+\varphi_c\, x_c\right)\cdot\theta_f\, , if\ x_f\geq\varphi_c \\ \left(\varphi_c-\dfrac{x_f\left(1-\varphi_f\right)}{\varphi_c}\right)\cdot\theta_c+\varphi_f\cdot x_f\cdot\theta_f\, , otherwise \end{cases}$$

***RC1 #4: The evaluation of the model fitting is limited because it is restricted to the root mean square error with a complementary analysis of the absolute deviation. Absolute deviation scaled by the measured water content would have been more relevant and could have been implemented in the parameter fitting procedure.***

We are thankful for the comment and conclude that the evaluation procedure was not described clearly enough. Regarding the methodology, we would like to highlight that the water retention measurements of each mixture were first represented with parametric water retention curve models (which we specified in the appendix). For the fitting procedure, the RMSE is used as diagnostics to assess the quality of fit of the hydraulic models to the retention data. In the fitting procedure, we minimize the sum of squared residuals, not the absolute deviation. This is common practice.

Of all parametric representations, the fitted curves for the pure components (and the intermediate mixture for the extended model scheme CM2) have then been used as model input for predicting the water retention characteristics of the binary mixtures. The curves fitted to the measured data of the mixtures are considered as reference curves. Finally, we compare the predicted curves with the reference curves and use both the RMSE and absolute deviation to evaluate the quality of the model predictions.

We will clarify the terminology in the revised manuscript: i) fitted functions for prediction ('fit4pred'), ii) predicted functions using the compositional models ('pred'), and iii) fitted functions to the measured data of the mixtures as reference for the predictions ('fit4ref') (see also our response to RC1#4). Furthermore, we will improve the structure of the Material & Methods section as outlined in the reply RC2 #4.

***RC1 #5: More information should be provided on parameter correlation and parameter uncertainty related to the estimation. Moreover, the fitted models are biased; the match for high water contents values is different than the match for the low water contents values (fig. 8)***

We believe that this comment relates to a misunderstanding (compare explanation in RC1 #4) as we present the comparison between reference curves ('fit4ref') and predicted curves ('pred') in Figure 8.

Here, our model predictions indeed perform better in the intermediate and less accurately in the wet range, which we discuss based on the absolute deviations. This result is not surprising though, since the retention characteristics close to saturation are highly influenced by soil structure, a hardly predictable quantity. We will point on soil structure in the discussion.

Parameter uncertainty and parameter correlation can only be quantified for the single curve fittings. This information is to our understanding not of high importance in this study, since only the shape of retention curves as a whole are analysed, not the values of the single parameters. We, refer, however, to Table A1-A4, where the parameter values are given. The 95% confidence intervals and the parameter correlations matrices could be added in the supplemental materials.

***RC1 #6: Discussion in section 3.6 has to be improved by addressing saturated hydraulic conductivity.***

See our reply to RC1#2.

*RC1 #7: Typo L40, L76, L80, L200, L310, L232 ???: Is it Technsol or Technosol or both ?*

Thank you for the remark. The exact terminology is, of course, Technosol. We apologize for the typos, which will be corrected accordingly.

*RC1 #8: Figure 4, 5, … have to be improved. They are too small.*

We thank the reviewer for addressing this issue and will increase the figure sizes and text size of the labelling accordingly.

*RC1 #9: Considering that the paper could be greatly improved by addressing also saturated hydraulic conductivity, that the discussion of the model and model fitting are not enough detailed, and that the topic is not at the heart of the HESS's themes, the paper should not be published in HESS. I encourage the authors to submit the paper after improvement to a more suitable journal.*

We thank the reviewer for the suggestions to improve our manuscript that we will consider, as stated in the replies above. To our opinion, especially the consideration of hydraulic conductivity (not only saturated) improved it greatly. We are convinced that the knowledge of the soil hydrology of constructed Technosols plays a key role in urban greening (green roofs, raised beds, facade greenery) and its services regarding urban water management (drought resilience, stormwater retention, grey water management).

---

## Author Comment (AC2)

*Reply to anonymous Reviewer #2*

*RC2 #1: The paper presents different ways to predict the water retention curve of binary mixtures (i.e., mixtures of two materials with different particle size distributions). The study was carried out within the context of the need to understand the hydraulic properties of constructed technosols that result from mixtures of geogenic materials of different particle size distributions. Willaredt et al. define two models and predict the water retention curves of different mixtures previously characterized in previous studies. The authors also fit the data to the bimodal water retention cure model, using an approach similar to Durner's (Durner, 1994). The manuscript is clearly organized ad relatively well-written. I reckon proofreading by a Native from English spoken countries. I have some concerns that should be addressed before potential publication:*

We thank the reviewer for her/his time and effort dedicated to substantially improving our manuscript and appreciate the positive feedback regarding the quality of the structure and language of the manuscript. We paid detailed attention to all comments and formulated concerns and will address them as follows. The manuscript was proofread by a native English speaker (see the acknowledgments) and the revised manuscript will be again proofread.

*RC2 #2: The topic of the paper lies partially in line with the area of hydrology. This study is clearly more geotechnical than hydrological. I join the feeling of the first reviewer concerning this aspect. I suggest strengthening the link with hydrological processes.*

To our opinion, this paper is rather hydrological than geotechnical. Regarding its relevance for hydrology, we agree that especially the consideration of hydraulic conductivity (not only saturated) will improve it greatly. We are convinced that the knowledge of the soil hydrology of constructed Technosols plays a key role in urban greening (green roofs, raised beds, facade greenery) and its services regarding urban water management (drought resilience, stormwater retention, grey water management) (see also reply RC1 #9). We will elaborate on this relevance more explicitly throughout the revised manuscript, e.g. by considering the unsaturated hydraulic conductivity as we describe in RC2 #3.

*RC2 #3: In the end, the authors propose an application of their model that is more related to hydrological sciences. They show how their model may be used to predict water content in technosoils for growing plants and trees. Using capillary models, they could increase the link with hydrology by predicting the unsaturated hydraulic conductivity from predicted water retention curves.*

We are very thankful for this comment, we will improve the manuscript by considering unsaturated hydraulic conductivity. Due to the lack of hydraulic conductivity observations for most data sets, we will apply the prediction scheme of complete unsaturated hydraulic conductivity functions, including the capillary bundle model, based on the theory developed by Peters et al. (2023) (HESSD https://hess.copernicus.org/preprints/hess-2022-431/). See also our reply to reviewer 1 (RC1 #2). This will enable modelling transport processes in constructed Technosols.

*RC2 #4: I suggest moving the appendix to the main text since it presents important mathematical aspects. This move would ease reading the result section, in which links to equations and models need to be more straightforward.*

We thank the reviewer for this suggestion. We make use of the parametric fitting models in this study rather as a means to represent the measured data for further processing. For a more detailed discussion on fitting water retention models to binary mixtures and a comparison of their performance, we propose to refer the reader to Willaredt and Nehls (2021). In this study, we aim to highlight the compositional model and therefore will leave the detailed equations of the fitted models in the supplementary material.
We agree, however, that the structure of the methodological section can be improved and will rearrange the Material & Methods section:

2. Materials & Methods

2.1 Compositional models

    2.1.1 Adapted Clarke model

    2.1.2 Basic scheme CM1

    2.1.3 Extended scheme CM2

2.2 Data sets of binary mixtures and their mathematical representation

    2.2.1 Data sets

    2.2.2 Mathematical representation

2.3 Testing

2.4 Model application

    2.4.1 Estimation of the distribution of water and air in constructed Technosols

    2.4.2 Prediction of hydraulic conductivity functions

Furthermore, we will clarify the terminology in the revised manuscript: i) fitted functions for prediction ('fit4pred'), ii) predicted functions using the compositional models ('pred'), and iii) fitted functions to the measured data of the mixtures as reference for the predictions ('fit4ref') (see also our response to RC1#4).

*RC2 #5: Several variables and models are only sometimes clearly presented in the appendix and the main text. Please, define all the variables and their related units.*

We apologize for this negligence and are thankful for this comment. We will define all variables and their units in the revised manuscript.

*RC2 #6: Regarding the optimization process, when fitted data are compared to observed data, the equations should be reminded, and the optimized parameter should be listed and discussed. No discussion of parameters is proposed.*

We believe that the models are clearly addressed. We do not want to pay too much attention to the values of the single parameters since only the shape of retention curves as a whole is of interest (see also our reply to reviewer 1 (RC1 #5)). We refer to the list of all optimized parameter values in the Appendix (Table A1-A4).

*RC2 #7: The physics should also be discussed. In several mixtures, the fine components have bimodal pore size distributions. Theoretically, this could lead to three modes for the mixture pore size distribution, with two for the fine part and one for the coarsest part. This aspect should be discussed.*

We are thankful for this comment and you are right that theoretically, we could achieve 3 modes. This will depend mostly on the difference in the maximum pore size of the single components and their homogeneity. It is certainly a very interesting and important topic. We will discuss this issue briefly in the revised manuscript: i) for the special case of mixing bricks with an inner pore-size distribution (PSD) mixed with other components, the inner pore-size distribution is not changed by the mixing. ii) Not every mode in the single component's PSD is necessarily visible in the mixtures because the PSD may intertwine in a way that cannot be described straight-forward on a physical basis.

*RC2 #8: The impact of the symmetrical and multimodal pore size distributions of components should be discussed concerning their impact on the bulk water retention of the mixture.*

We refer to our reply to RC2 #7.

*RC2 #9: More detailed comments: The authors will find a list of suggestions and comments in the enclosed pdf document.*

We appreciate all suggestions for editorial improvements by the reviewer given in the manuscript and will consider them in the revised version. We are also thankful for highlighting the strong points of our manuscript. The replies to the comments in the manuscript are given in the following.

*Title: I join reviewer 1 in his comment. Water retention is important, but additional impacts on the prediction of the saturated hydraulic conductivity could have been better.*

We agree and refer to the reply to RC2 #3. We will update the title to "Predicting Soil Hydraulic Properties for Binary Mixtures – Concept and Application for Constructed Technosols"

*l.11 the two modes of the …*

Here we refer to the pore-size distributions (PSD) of the single components and do not want to address the shape but only the maxima of the PSD. Regarding the consideration of the modality of the PSD we refer to RC2 #7 and RC2 #8.

*l.15: This conclusion seems to be more related to geotechnical issues than hydrological issues. Define here the targeted application.*

In RC2 #2 and RC2#3 we elaborate on the link with hydrology of this study and how we intend to strengthen it, e.g. by including the prediction of soil hydraulic conductivity.

In the abstract, the last sentence will be: "The knowledge of the soil hydraulic properties of any mixing ratio facilitates the choice of a Technosol composition that matches e.g. specific urban water management purposes and meets the required hydraulic properties to supply water to urban green."

*l.38: Please, give more details on these variables, or at least their names (like "bulk density" for "BD").*

Thank you for this indication. Will be done.

*l.40: and also the hydraulic conductivity*

Will be done.

*Figure 2: To my point, this figure is not explicit enough. You should avoid the 3D perspective and do a figure with a 2D perspective instead.*

We appreciate the assessment that Figure 2 can be improved and changed the opacity to reduce the appearance of a third dimension.

[Figure]

[Figure]

*l.101 a) why in italics?*

Thanks. The typo will be changed.

*l.101 b) not very clear sentence*

We acknowledge the reviewers' critic and will revise this section carefully (see also our reply to RC1 #3).

*l.110 This corresponds to the zero mixing model, right?*

Yes, indeed. We will state it directly in the following sentences: "This approach corresponds to the "zero-mixing" concept and is a weighted superposition of the WRCs of the two components to predict the […]"

*l.117 The input of this model is not very clear.*

Thank you for this remark, we will reformulate the description as follows:
"For the extended scheme of the compositional model, an additional WRC is required for predicting a mixtures' WRC. The additional WRC should represent a mixture of similar shares of both components. Therefore it is referred to as the WRC of an intermediate mixture $\theta_m$. We apologize for the inconsistency in our nomination. The variable $x_i$ will be replaced with $x_m$ and stands for the bulk volumetric share of component a in the intermediate mixture. We will update Equation 4 to:

$$\theta_{pred} = \begin{cases} \dfrac{x_a}{x_m}\theta_m + \left(1 - \dfrac{x_a}{x_m}\right)\theta_b \,, if \ x_a < x_m \\[3mm] \dfrac{1-x_a}{1-x_m}\theta_m + \left(1 - \dfrac{1-x_a}{1-x_m}\right)\theta_a \,, if \ x_a > x_m \end{cases}$$

*l.140 Could you define xi,m above and elaborate more on this (even though this is quite straightforward)?*

We appreciate this remark, the variables $x_{i,m}$ and $x_{i,v}$ will be clearly defined in the revised manuscript.

*l.157 I reckon to put the appendix here, to have the whole information in the same place.*

We refer to the reply to RC2 #4.

*l.158 We need the equations of the water retention functions. Please, add equations in the appendix here.*

We refer to the reply to RC2 #4.

*l.166 The type of models (uni-modal versus bi-modal) should be considered based on physical considerations instead of the number of points.*

The reviewer is certainly right, that the physical structure of some components leads to multi-modal PSD, however, such PSD can simply not be detected with a very limited number of data points.
Especially for a small number of data points, fitting a small number of parameters results in more robust fitting and consequently in more robust prediction. We refer to the data of Willaredt and Nehls (2021), which contain the information for more complex retention functions.

*l.167 It would help if you gave the equations to allow the reader to understand the differences. I know the constraint (m=1-1/n), but that needs to be mentioned clearly, and the reader may need clarification.*

We will refer to equation A3 in the revised manuscript. This will make it clear.

*l. 184 Not clear. Why this value? In addition, do you consider this value at one boundary of the profile and then compute over the whole profile (considering hydrostatic water pressure head).*

We apologize that we forgot to clearly define the underlying assumptions. At the lower boundary we assumed full saturation and hydrostatic equilibrium for the matric head distribution in the container. Thus, pF 1.7 will occur at the top of the container. We will revise the manuscript accordingly.

*l.214 Unclear, please, elaborate*

see reply to comment on L 166.

*Figure 4. For the line "fit", please remind the corresponding equation (or model).*

Thanks for that comment. We will refer to the according equation given in the appendix.

*l.238 ??*

The sentence could be improved to "For constructed Technosols containing coarse particles with inner porosity, the Clarke model could be applied in a modified version, that accounts for additional water retention within the coarse particles."

*l.249 Which one? Again, remain the corresponding model (I guess PDI).*

We thank the reviewer for this remark and suggest specifying the model by changing the sentence as follows: "The deviations here reflect the comparably poor fit of the uni-modal constrained van Genuchten model used to represent the data of the pure peat (RMSE 0.029 $m^3m^{-3}$ )."

*Figure 6: You should provide the parameter estimate for the fitted data. These may be discussed and give further ideas about the activation of porosity.*

Regarding the parameter estimates we refer to Table A1-A4 in the Appendix, where we provide the values of each parameter together with the RMSEs describing the fitting quality for each mixture. Regarding the activation of porosity, we, unfortunately, do not understand the meaning.

*Figure 8: Which model? PDI model? Give details here.*

We refer to the reply to RC2 #4 and suggest accordingly replacing the terms "observed and fitted water contents" with "reference water contents" and "the parametric fittings" with "reference curves".

*l. 283 "for the CM1 model, "*

We appreciate the suggestion and will include it in the revised manuscript.

*l. 284 "deviations, apart from a few exeptions". Then, you need to list them.*

We appreciate the suggestion and intend to improve these sentences to "The extended model approach leads to smaller RMSE and also to smaller absolute deviations, apart from a few exceptions: in the wet range for the mixture C7B3 of the data set by Willaredt and Nehls (2021) and in the medium to dry range for the mixture C2E8 of the data set by Deeb et al. (2016)"

*Figure 9. Be more precise. Where is this value applied? At the top of the container?*

We suggest addressing this remark by changing the sentence to: "Distribution of volumetric water and air content over different depths at hydrostatic equilibrium in a container (corresponding to pF = 1.7 at the top of the container) [...]" see also our reply to your comment in l. 184.

*l.308 How can we change this unphysical feature?*

You are right, that is unphysical. The preparation method of the samples in the study of Deeb et al. (2016) led to rather high deviations between the bulk densities of the replicates. We calculated the porosity of their mixtures therefore based on a mean bulk density. We will define the porosity as $\max(\theta_s, 1-BD/PD)$, this will avoid negative air contents.

*l.318 Do you refer to the CM2 model when you state this?*

We thank the reviewer for this remark and propose to express the conclusion more precise by reformulating the sentence to:
"The introduced compositional model approach, in the basic, as well as, extended scheme, was shown to be applicable to mixtures of components characterized by a small difference between their pore space distribution maxima ($\Delta PSD_{max}$). It can be concluded that the compositional model approach performs best, based on water [...]"

---

## Author Response (AR2)

*Manuscript ID: hess-2022-265*

*Final Author Comment*

Dear Philippe Ackerer,

thank you for being the handling Editor for our manuscript. We appreciate the constructive feedback and comments of the three anonymous reviewers, which have improved our manuscript. In the revised and technically corrected manuscript, all additions have been highlighted in blue, while the text we removed is highlighted in red. Please find below our the point-by-point reply to the reviewer comments. We list all relevant changes we have made in the revised manuscript integrating our replies to the comments of the reviewers. The indicated line numbering refers to the Authors track-changes file.

Additionally, we would like to inform you that after the revision process, Thomas Nehls suggested to adjust the order of authors, as their contributions have shifted. We all agreed on the following order:
Moreen Willaredt, Thomas Nehls, and Andre Peters.

***Point-by-point reply to the comments***
***Anonymous reviewer 1***

**RC1 #1: The motivation and objectives of this work are clearly described.**

We thank for the dedicated work and time to review our manuscript and appreciate the positive feedback regarding our motivation and objectives.

**RC1 #2: Compared to the usual high scientific level of papers published in HESS, the paper should be improved by addressing also the saturated hydraulic conductivity.**

We agree with the reviewer that considering the hydraulic conductivity (not only saturated) does substantially improve the manuscript, enhances the scope and highlights the indented hydrological focus of our study. We are aware of the particular importance of soil hydraulic properties for numerical simulations of flow and transport processes in the soil-plant-atmosphere system. In the revised manuscript we have used the theory developed by Peters et al. (2023) (HESSD https://hess.copernicus.org/preprints/hess-2022-431/), to predict hydraulic conductivity curves (HCCs) which does not require any conductivity measurements because these data is missing for most of the used data sets.

Addressing the hydraulic conductivity, we have added the section "2.4.1 Prediction of hydraulic conductivity functions", where we describe the methodology (line 229 f) for predicting HCCs for two data sets of our study and the determination of the saturated matrix conductivity $K_{s,matrix}$ [cm d$^{-1}$] for each mixture. We have further added the section "3.6.1 Hydraulic conductivity prediction" (line 383 f), that presents and discusses the results of our predictions.
The extended scope of the revised manuscript is represented in the Title, that we have changed to be "Predicting Soil Hydraulic Properties for Binary Mixtures - Concept and Application for Constructed Technosols". Throughout the manuscript we have adjusted the focus from water retention curves (WRCs) to soil hydraulic properties (SHP) (i.e. line 3 f, 18, 84, 434). Furthermore, we have incorporated the hydraulic conductivity into the relevant sections (line 10 f, 46, 88, 432).

*RC1 #3: It should also provide a deeper discussion of the model parameter xcrit whose impact on the model results remains unclear to me.*

We agree with the reviewer, that the model parameter $x_{crit}$ is not discussed deeply and are thankful for that comment. It reveals the potential to improve the formulation of our adapted Clarke model by reviewing the parameter $x_{crit}$. The parameter $x_{crit}$ originates from the work on porosity in binary mixtures of coarse and fine particles of Clarke (1979) and refers to a critical volumetric fraction of fine particles. According to his concept (confirmed by later studies e.g. Zhang et al. (2011) and Sakaki and Smits (2015)), $x_{crit}$ corresponds to the porosity of the coarse fraction. Therefore, introducing the extra parameter $x_{crit}$ is not required and we suggest simplifying the model by replacing it now with $\varphi_c$, which is the porosity of the coarse fraction. This critical content can be understood as a threshold at which the relation between the porosity and the volumetric fraction of the fine particles changes from "coarse-controlled" to "fine-controlled" (Sakaki and Smits, 2016). These two regimes are equally observable in the water retention data of binary mixtures of coarse and fine particles (Sakaki and Smits, 2016). We accordingly implemented these two regimes in our adapted Clarke model by providing two equations for each respective case.

In the revised manuscript we have added the explanation in line 109:

"The volumetric share of the fine fraction $x_f$ [-] in the mixture delineates the two regimes. The threshold, at which the relation between the porosity and the volumetric fraction of the fine component changes between the regimes, corresponds to $x_f = \varphi_c$ (Sakaki and Smits, 2015), where $\varphi_c$ [-] stands for the porosity in the coarse component of the binary mixture."

and have replaced the parameter $x_{crit}$ in Eq. 2 with the porosity of the coarse fraction $\varphi_c$ [-] leading to:

$$\theta_{pred} = \begin{cases} \left( x_f + \varphi_c\, x_c \right) \cdot \theta_f, & if\ x_f \geq \varphi_c \\ \left( \varphi_c - \dfrac{x_f\left(1 - \varphi_f\right)}{\varphi_c} \right) \cdot \theta_c + \varphi_f \cdot x_f \cdot \theta_f, & otherwise \end{cases}$$

*RC1 #4: The evaluation of the model fitting is limited because it is restricted to the root mean square error with a complementary analysis of the absolute deviation. Absolute deviation scaled by the measured water content would have been more relevant and could have been implemented in the parameter fitting procedure.*

We are thankful for the comment and conclude that the evaluation procedure was not described clearly enough. Regarding the methodology, we would like to highlight that the water retention measurements of each mixture were first represented with parametric water retention curve models (which we specified in the appendix). For the fitting procedure, the RMSE is used as diagnostics to assess the quality of fit of the hydraulic models to the retention data. In the fitting procedure, we minimize the sum of squared residuals, not the absolute deviation. This is common practice.

Of all parametric representations, the fitted curves for the pure components (and the intermediate mixture for the extended model scheme CM2) have then been used as model input for predicting the water retention characteristics of the binary mixtures. The curves fitted to the measured data of the mixtures are considered as reference curves. Finally, we compare the predicted curves with the reference curves and use both the RMSE and absolute deviation to evaluate the quality of the model predictions.

In the revised manuscript we have improved the terminology to clarify the evaluation procedure: i) fitted functions for prediction ('fit4pred'), ii) predicted functions using the compositional models ('pred'), and iii) fitted functions to the measured data of the mixtures as reference for the predictions ('fit4ref') (see also our response to RC1#4). We have implemented the new terminology consistently throughout the text, including in equations and figures (i.e Fig. 4-7 and Fig.10), and added an explanatory sentence in line 210 f:

> "The fitted curves for the pure components and the intermediate mixtures (referred to as „fit4pred") were used as model input to predict the water retention curves (referred to as „pred") of all binary mixtures. The fitted curves for all other mixtures were used as reference curves (these are referred to as „fit4ref") to subsequently assess the quality of predictions."

Further, we have restructured the Material & Methods section to better reflect the methodology, as described in our reply to RC2 #4.

*RC1 #5: More information should be provided on parameter correlation and parameter uncertainty related to the estimation. Moreover, the fitted models are biased; the match for high water contents values is different than the match for the low water contents values (fig. 8)*

We believe that this comment relates to a misunderstanding (compare explanation in our reply to RC1 #4) as we present the comparison between reference curves ('fit4ref') and predicted curves ('pred') in Figure 8. Here, our model predictions indeed perform better in the intermediate and less accurately in the wet range, which we discuss based on the absolute deviations. This result is not surprising though, since the retention characteristics close to saturation are highly influenced by soil structure, a hardly predictable quantity. We have added a sentence pointing on soil structure in the discussion (lines 331f):

> "That is not surprising, since the retention characteristics close to saturation are highly influenced by soil structure and thus hardly predictable"

Parameter uncertainty and parameter correlation can only be quantified for the single curve fittings. This information is to our understanding not of high importance in this study, since only the shape of retention curves as a whole are analysed, not the values of the single parameters. We, refer, however, to Table A1-A4, where the parameter values are given. We have decided against including the 95% confidence intervals and the parameter correlations matrices in the supplemental materials as we believe that their inclusion may divert attention from the main focus of our study, which is the development of a compositional model approach for predicting water retention curves.

*RC1 #6: Discussion in section 3.6 has to be improved by addressing saturated hydraulic conductivity.*

Please see our reply to RC1#2. We have now added a new subsection titled "3.6.1 Hydraulic conductivity prediction" to the Results & Discussion section. The subsection presents the predictions of the HCCs obtained with the approach developed by Peters et al. (2023).

*RC1 #7: Typo L40, L76, L80, L200, L310, L232 ???: Is it Technsol or Technosol or both ?*

Thank you for the remark. The exact terminology is, of course, Technosol. We apologize for the typos, which we have corrected accordingly.

*RC1 #8: Figure 4, 5, … have to be improved. They are too small.*

We thank the reviewer for addressing this issue and have improved the figure sizes and text size of the labeling accordingly. We have reiterated the figures and changed the line coloring to distinguish between predicted (blue) and fitted (black) WRCs. Additionally, we have changed the legends and captions with the updated more straight-forward terminology (see reply to RC1 #4).

*RC1 #9: Considering that the paper could be greatly improved by addressing also saturated hydraulic conductivity, that the discussion of the model and model fitting are not enough detailed, and that the topic is not at the heart of the HESS's themes, the paper should not be published in HESS. I encourage the authors to submit the paper after improvement to a more suitable journal.*

We thank the reviewer for the suggestions to improve our manuscript that we have considered, as stated in the replies above. To our opinion, especially the consideration of hydraulic conductivity (not only saturated) has improved it greatly. We are convinced that the knowledge of the soil hydrology of constructed Technosols plays a key role in urban greening (green roofs, raised beds, facade greenery) and its services regarding urban water management (drought resilience, stormwater retention, grey water management). We have highlighted the relevance of our manuscript for hydrology in the Conclusions by adding the paragraph: (lines 434f):

> "The knowledge of the soil hydraulic properties of any mixing ratio enables the quick choice of a binary Technosol composition, based on either estimated air capacity, wilting point capacity and available water capacity or the modelled water balance of a soil-plant-atmosphere system e.g. in urban green infrastructure. Through this, planning for efficient water management in urban green infrastructure dedicated to different purposes (e.g. rainwater, grey water, irrigation etc.), is made possible"

**Anonymous reviewer 2**

*RC2 #1: The paper presents different ways to predict the water retention curve of binary mixtures (i.e., mixtures of two materials with different particle size distributions). The study was carried out within the context of the need to understand the hydraulic properties of constructed technosols that result from mixtures of geogenic materials of different particle size distributions. Willaredt et al. define two models and predict the water retention curves of different mixtures previously characterized in previous studies. The authors also fit the data to the bimodal water retention cure model, using an approach similar to Durner's (Durner, 1994). The manuscript is clearly organized ad relatively well-written. I reckon proofreading by a Native from English spoken countries. I have some concerns that should be addressed before potential publication:*

We thank the reviewer for her/his time and effort dedicated to substantially improving our manuscript and appreciate the positive feedback regarding the quality of the structure and language of the manuscript. We paid detailed attention to all comments and formulated concerns and addressed them as follows. The manuscript was proofread by a native English speaker (see the acknowledgments) and the revised manuscript has been proofread again and improved accordingly.

*RC2 #2: The topic of the paper lies partially in line with the area of hydrology. This study is clearly more geotechnical than hydrological. I join the feeling of the first reviewer concerning this aspect. I suggest strengthening the link with hydrological processes.*

To our opinion, this paper is rather hydrological than geotechnical. Regarding its relevance for hydrology, we agree that especially the consideration of hydraulic conductivity (not only saturated) has improved it greatly. We are convinced that the knowledge of the soil hydrology of constructed Technosols plays a key role in urban greening (green roofs, raised beds, facade greenery) and its services regarding urban water management (drought resilience, stormwater retention, grey water management) (see also reply RC1 #9). We have elaborated on this relevance more explicitly in the revised manuscript, e.g. by considering the unsaturated hydraulic conductivity (see our reply to RC1#2 and RC2 #3 for details) and by adding a paragraph in the Conclusion (lines 434f):

> "The knowledge of the soil hydraulic properties of any mixing ratio enables the quick choice of a binary Technosol composition, based on either estimated air capacity, wilting point capacity and available water capacity or the modelled water balance of a soil-plant-atmosphere system e.g. in urban green infrastructure. Through this, planning for efficient water management in urban green infrastructure dedicated to different purposes (e.g. rainwater, grey water, irrigation etc.), is made possible."

*RC2 #3: In the end, the authors propose an application of their model that is more related to hydrological sciences. They show how their model may be used to predict water content in technosoils for growing plants and trees. Using capillary models, they could increase the link with hydrology by predicting the unsaturated hydraulic conductivity from predicted water retention curves.*

We are very thankful for this comment and have improved the manuscript by integrating the unsaturated hydraulic conductivity. Due to the lack of hydraulic conductivity observations for most data sets, we have applied the prediction scheme of complete unsaturated hydraulic conductivity functions, including the capillary bundle model, based on the theory developed by Peters et al. (2023) (HESSD https://hess.copernicus.org/preprints/hess-2022-431/). This enhancement enables modelling transport processes in constructed Technosols (see also reply to RC1#2).

We have added the section "2.4.1 Prediction of hydraulic conductivity functions", where we describe the methodology (lines 229 f) for predicting HCCs for two data sets of our study and the determination of the saturated matrix conductivity $K_{s,matrix}$ [cm d$^{-1}$] for each mixture. We have further added the section "3.6.1 Hydraulic conductivity prediction" (lines 383 f), that presents and discusses the results of our predictions. The extended scope of the revised manuscript is as well represented in the Title, that we have updated to be:

> "Predicting Soil Hydraulic Properties for Binary Mixtures - Concept and Application for Constructed Technosols".

Throughout the manuscript we have adjusted the focus from water retention curves (WRCs) to soil hydraulic properties (SHP) (i.e. line 3 f, 18, 84, 434). Furthermore, we have incorporated the hydraulic conductivity into the relevant sections (line 10 f, 46, 88, 432).

*RC2 #4: I suggest moving the appendix to the main text since it presents important mathematical aspects. This move would ease reading the result section, in which links to equations and models need to be more straightforward.*

We thank the reviewer for this suggestion. We make use of the parametric fitting models in this study rather as a means to represent the measured data for further processing. For a more detailed discussion on fitting water retention models to binary mixtures and a comparison of their performance, we propose to refer the reader to Willaredt and Nehls (2021). In this study, we aim to highlight the development of the compositional model and therefore leave the detailed equations of the fitted models in the supplementary material. We have improved the section "Mathematical representation" by consistently referencing to the respective equations elaborated in the Appendix (lines 200, 203, 208 and 209).
We agree, however, that the structure of the methodological section can be improved and have rearranged the Material & Methods section to:

   2. Materials & Methods

   2.1 Concept of compositional models

      2.1.1 Adapted Clarke model

      2.1.2 Basic scheme CM1

      2.1.3 Extended scheme CM2

   2.2 Data sets of binary mixtures and their mathematical representation

      2.2.1 Data sets

      2.2.2 Mathematical representation

   2.3 Testing

   2.4 Model application

      2.4.1 Prediction of hydraulic conductivity functions

      2.4.2 A case study of predicted water and air distribution

Furthermore, we have clarified the terminology in the revised manuscript: i) fitted functions for prediction ('fit4pred'), ii) predicted functions using the compositional models ('pred'), and iii) fitted functions to the measured data of the mixtures as reference for the predictions ('fit4ref') (see also our response to RC1#4).

*RC2 #5: Several variables and models are only sometimes clearly presented in the appendix and the main text. Please, define all the variables and their related units.*

We apologize for this negligence and are thankful for this comment. We have defined all variables and their units in the revised manuscript. (lines 16, 99, 112f, 175, 255f, 458f, 461, 473)

*RC2 #6: Regarding the optimization process, when fitted data are compared to observed data, the equations should be reminded, and the optimized parameter should be listed and discussed. No discussion of parameters is proposed.*

We believe that the models are clearly addressed. We do not want to pay too much attention to the values of the single parameters since only the shape of retention curves as a whole is of interest (see also our reply to

RC1 #5). In the improved manuscript we have included references to the Tables listing all optimized parameter values in the Appendix in line 196:

> "The detailed model descriptions and the obtained parameters together with the RMSE between the models and observations are summarized in the Appendix (Tab. A1-A4)."

Further, we have included consistent referencing to the respective equations and tables in the Appendix in the sections Methodology (lines 200, 203, 208 and 209) and Results & Discussion (lines 314, 320, 223 and within the captions of Fig 4-7).

*RC2 #7: The physics should also be discussed. In several mixtures, the fine components have bimodal pore size distributions. Theoretically, this could lead to three modes for the mixture pore size distribution, with two for the fine part and one for the coarsest part. This aspect should be discussed.*

We are thankful for this comment and you are right that theoretically, we could achieve 3 modes. This will depend mostly on the difference in the maximum pore size of the single components and their homogeneity. It is certainly a very interesting and important topic. We have added a paragraph discussing this issue in line 278f:

> "In mixtures formulated with more then two components, or with components containing coarse particles with inner porosity (e.g. bricks), three maxima would have to be considered. Not every mode in the single component's PSD is necessarily visible in the mixtures because the PSD may intertwine."

*RC2 #8: The impact of the symmetrical and multimodal pore size distributions of components should be discussed concerning their impact on the bulk water retention of the mixture.*

We refer to our reply to RC2 #7.

*RC2 #9: More detailed comments: The authors will find a list of suggestions and comments in the enclosed pdf document.*

We appreciate all suggestions for typos and editorial improvements by the reviewer given in the manuscript and have comprehensively considered them in the revised version. We are also thankful for highlighting the strong points of our manuscript. The replies to the comments in the manuscript and how they are addressed in the revised version are given in the following.

*Title: I join reviewer 1 in his comment. Water retention is important, but additional impacts on the prediction of the saturated hydraulic conductivity could have been better.*

We agree and refer to the reply to RC2 #3. We have updated the title to:

> "Predicting Soil Hydraulic Properties for Binary Mixtures – Concept and Application for Constructed Technosols"

**l.11 the two modes of the …**

Here we refer to the pore-size distributions (PSD) of the single components and do not want to address the shape but only the maxima of the PSD. Regarding the consideration of the modality of the PSD we refer to RC2 #7 and RC2 #8.

**l.15: This conclusion seems to be more related to geotechnical issues than hydrological issues. Define here the targeted application.**

In our reply to RC2 #2 and RC2#3 we elaborate on the link with hydrology of this study and how we strengthen it in the revised manuscript, e.g. by including the prediction of soil hydraulic conductivity curves. In the abstract, we have changed the last sentence accordingly to (line 18):

> "The prediction of the soil hydraulic properties of any mixing ratio facilitates the simulation of flow and transport processes in constructed Technosols before they are produced e.g. for specific urban water management purposes."

**l.38: Please, give more details on these variables, or at least their names (like "bulk density" for "BD").**

Thank you for this indication. We have changed the listing of the variables in the sentence using their names instead (line 37f):

> "Using dose-response curves they were able to describe six basic soil properties, which are important for agricultural use: total C, available phosphorus, cation exchange capacity, pH in water, the water content at a pressure head of $h = -100$ cm and the bulk density."

**l.40: and also the hydraulic conductivity**

We have added the hydraulic conductivity to the enumeration of characteristics (line 48).

**Figure 2: To my point, this figure is not explicit enough. You should avoid the 3D perspective and do a figure with a 2D perspective instead.**

We appreciate the assessment that Figure 2 can be improved and have changed the opacity to reduce the appearance of a third dimension.

[Figure]

[Figure]

**l.101 a) why in italics?**

Thanks. The typo has been changed.

We acknowledge the reviewers' critic and revised this section carefully (see our reply to RC1 #3 for the improvement of Equation 2). Further, we have changed the not very clear description to:

> "The volumetric share of fine particles is effectively larger in ideally mixed and fine-controlled mixtures ($x_f \geq \varphi_c$ ) compared to the bulk volumetric content of fine particles. The difference corresponds to the porosity in the bulk volumetric share of coarse particles as this volume is filled by fine particles."

*l.110 This corresponds to the zero mixing model, right?*

Yes, indeed. We have stated it in the revised manuscript directly in the following sentences (line 142):

> "This approach corresponds to the „zero-mixing" concept and is a weighted superposition of the WRCs of the two components to predict the WRC of the [...]"

Thank you for this remark, we reformulated the description of the model input (lines 150f):

> "For the extended scheme of the compositional model, an additional WRC is required for predicting a mixture's WRC. The additional WRC should represent a mixture of similar shares of both components. Therefore it is referred to as the WRC of an intermediate mixture $x_m$ [-] (intermediate mixing concept in Fig. 2). The motivation behind the extended scheme is to analyse if a slight increase in measurement effort leads to more sound predictions."

We apologize for the inconsistency in our nomination. We have replaced the variable $x_i$ with $x_m$ , standing for the bulk volumetric share of component a in the intermediate mixture. We have updated Equation 4 to:

$$\theta_{pred} = \begin{cases} \dfrac{x_a}{x_m}\theta_m + \left(1 - \dfrac{x_a}{x_m}\right)\theta_b \, , if \; x_a < x_m \\[3ex] \dfrac{1-x_a}{1-x_m}\theta_m + \left(1 - \dfrac{1-x_a}{1-x_m}\right)\theta_a \, , if \; x_a > x_m \end{cases}$$

*l.140 Could you define xi,m above and elaborate more on this (even though this is quite straightforward)?*

We appreciate this remark, the variables $x_{i,m}$ and $x_{i,v}$ are now clearly defined in the revised manuscript (lines 177f):

> "Walczak et al. (2002) created mixtures of peat and sand with mass specific contents of dry peat $x_{i,m}$ of 0, 0.05, 0.2, 0.4, 0.6, 0.8 and 1 (mass/mass), with i [-] referring to the specific mixture. For our analysis the volumetric peat content $x_{i,v}$ [-] of each mixture was determined based on the given bulk densities ($BD_{meas}$ ) and the [...]"

*l.157 I reckon to put the appendix here, to have the whole information in the same place.*

We have improved the section "Mathematical representation" by consistently referencing to the respective equations elaborated in the Appendix (lines 200, 203, 208 and 209) and we refer to the reply to RC2 #4.

*l.158 We need the equations of the water retention functions. Please, add equations in the appendix here.*

We have improved the section "Mathematical representation" by consistently referencing to the respective equations elaborated in the Appendix (lines 200, 203, 208 and 209) and we refer to the reply to RC2 #4.

*l.166 The type of models (uni-modal versus bi-modal) should be considered based on physical considerations instead of the number of points.*

The reviewer is certainly right, that the physical structure of some components leads to multi-modal PSD, however, such PSD can simply not be detected with a very limited number of data points. Especially for a small number of data points, fitting a small number of parameters results in more robust fitting and consequently in more robust prediction. We refer to the data of Willaredt and Nehls (2021), which contain the information for more complex retention functions. We have included the description on the relation between model choice and number of observations in the sentence (line 218f):

> "For those data sets unimodal models were applied, as the fitting of a small number of parameters results in more robust fitting and consequently more robust predictions."

*l.167 It would help if you gave the equations to allow the reader to understand the differences. I know the constraint (m=1-1/n), but that needs to be mentioned clearly, and the reader may need clarification.*

We have included a reference to equation A3 in the revised manuscript. This will make it clear.

*l. 184 Not clear. Why this value? In addition, do you consider this value at one boundary of the profile and then compute over the whole profile (considering hydrostatic water pressure head).*

We apologize that we forgot to clearly define the underlying assumptions. At the lower boundary we assumed full saturation and hydrostatic equilibrium for the matric head distribution in the container. Thus, pF 1.7 will occur at the top of the container. In the revised the manuscript we have included a description of the boundary conditions in line 257f:

> "As an example we chose a 0.5 m high raised bed with constant water saturation at the bottom. We furthermore assume hydrostatic equilibrium and calculate the matric potential across the whole profile, thus the matric potential at the upper boundary is approximately pF 1.7."

*l.214 Unclear, please, elaborate*

Please see our reply to comment on L 166.

*Figure 4. For the line "fit", please remind the corresponding equation (or model).*

Thanks for the feedback. We have updated the legends of Fig. 4 – 7 with the terminology we elaborated in RC2#4. Additionally, we have included references in the figures' captions to the Tables in the Appendix which specify the fitting models used for mathematical representation and list the corresponding parameters.

*l.238 ??*

The sentence has been improved to (lines 327):

> "For constructed Technosols, that contain coarse fragments with inner porosity, a modified version of the Clarke model that accounts for water retention within the coarse particles, could be applied."

*l.249 Which one? Again, remain the corresponding model (I guess PDI).*

We thank the reviewer for this remark and have specified the model in the revised manuscript (line 341):

> "The deviations here reflect the comparably poor fit of the original unimodal constrained model of van Genuchten (1980) used to mathematically represent the data of the pure peat (RMSE 0.029 $m^3 m^{-3}$ )."

*Figure 6: You should provide the parameter estimate for the fitted data. These may be discussed and give further ideas about the activation of porosity.*

Regarding the parameter estimates we refer to Table A1-A4 in the Appendix, where we provide the values of each parameter together with the RMSEs describing the fitting quality for each mixture. Regarding the activation of porosity, we, unfortunately, do not understand the meaning.

*Figure 8: Which model? PDI model? Give details here.*

We refer to improved terminology we have elaborated in the reply to RC2 #4 and have updated the terminology in the figure caption accordingly.

*l. 283 "for the CM1 model, "*

We appreciate the suggestion and have included it in the revised manuscript.

*l. 284 "deviations, apart from a few exeptions". Then, you need to list them.*

We appreciate the suggestion and have improved these sentences to (lines 389f):

> "The extended scheme CM2 leads to smaller RMSE and also to smaller absolute deviations, except for a few cases: in the wet range for the mixture C7B3 of the data set by Willaredt and Nehls (2021) and in the medium to dry range for the mixture C2E8 of the data set by Deeb et al. (2016)."

*Figure 9. Be more precise. Where is this value applied? At the top of the container?*

We have addressed this remark in our reply to the comment on l. 184.

*l.308 How can we change this unphysical feature?*

You are right, that is unphysical. The preparation method of the samples in the study of Deeb et al. (2016) led to rather high deviations between the bulk densities of the replicates. We calculated the porosity of their mixtures therefore based on a mean bulk density. In the revised manuscript we have changed this approach in the revised manuscript and instead calculate the air content as the difference between the saturated water content $\theta_s$ and the water content at the respective soil depth in the container $\theta(L)$ (line 214):

> "The air content is simply calculated as $\theta_s - \theta(z)$ , where $\theta_s$ [$m^3 m^{-3}$ ] stands for the water content at saturation and $\theta(z)$ [$m^3 m^{-3}$ ] stands for the water content at the matric potential corresponding to the soil depth z [m] in the container."

*l.318 Do you refer to the CM2 model when you state this?*

We thank the reviewer for this remark and have expressed it in the conclusion more precisely by reformulating the sentence to:

> "The introduced compositional model approach, in the basic as well as extended scheme, was shown to be applicable to mixtures of components characterised by a small difference in their pore space distribution maxima ($\Delta PSD_{max}$). It can be concluded that the compositional model approach performs best, based on water […]"

*Point-by-point reply to the comments for technical corrections after review*
*Anonymous reviewer 3*

*l. 7: misspelling : replace "it's" with "its".*

We appreciate this remark and have corrected it.

*l. 47: the link why the manuscript's contribution is essential to the readership of HESS could be elaborated a bit.*

We appreciate the suggestion and refer to our reply to RC1 #9 and RC2 #2, where we elaborate on why the manuscript's contribution is essential to the readership of HESS. Further, we added the sentence in the suggested line:

> "Nevertheless, the planning for efficient water management in urban green infrastructure requires the knowledge of the soil hydraulic properties of the used Technosols."

*l. 101: "theta pred" should be referred to throughout as "predicted volumetric water content at any matric potential", rather than "predicted water retention curve".*

We thank the reviewer for the constructive suggestions. In this case, we prefer to keep our chosen, and commonly used terminology "retention curve". We believe that it contributes to the clarity of our methodology (see also our reply to RC2 #4).

*Table 1: definition of BD ("bulk density") and PD ("particle density") is missing.*

We appreciate this remark and have completed the definitions.

*l. 135: Table 2 is listed before Table 1 in the text. Please change.*

We appreciate this remark and have corrected it.

*l. 146: Unclear how a matrix potential of over 10m could be physically achieved.*

We thank the reviewer for this remark and refer to the original research articles of Deeb et al. (2016) and Walczak et al. (2002) for the available details.

*l.147 - 155: Text is not easy to understand. Too many details for the reader!*

We thank the reviewer for this feedback. However, we consider this conversion step essential - especially for guaranteeing reproducibility of our work - and therefore prefer keeping the detailed description in the text.

*l. 158: The designation of the soil mixtures C0B10, C2B8, etc. is not easy to understand. What is the purpose of the addition "B10", "B8", etc.? The same question arises for the reader for the designation of the soil samples listed in Table 2 ("S10", "S8", etc.). A consistently simpler designation would be desirable!*

Thanks for this feedback. The chosen designation means to highlight the fact, that the mixtures are binary combinations of two components, ranging from volumetric shares of the pure first component (100/0) to volumetric shares of the pure second component (0/100). Hence, they represent a full range of possible mixtures. We understand the point that representing the percentages (e.g. C20B80) would be more straightforward. However, we opted for the shorter version C2B8 for reasons of readability and clarity, especially in the figures and tables. Above all, we used the similar designation in a previous paper and chose to stick with it for continuity.

*Figure 3: "dtheta" used for the y-axis is not explained.*

We thank the editor for this hint and completed the figure caption accordingly. We decided to include the definition of $r_{eff}$ from the x a-axis for completeness:

"[…] $d\theta$ stands for the pore density and $r_{eff}$ for the effective pore radius."

*Figure 5: Its size should be increased. Replace "gray" by "grey". Idem for sizes of figures 6 and 7.*

We appreciate this remark and have replaced "gray" by "grey". The maximum size of the figures is given by the journals instructions and we had optimized the panel and label sizes for the maximum size and for clarity.

*Table 3: Which model was used to obtain the minimum and maximum deviations listed here?*

We thank the reviewer for this indication and included the respective model in the caption as follows:

"Maximum and minimum deviation between observations and the corresponding mathematical representations of volumetric water contents of all observed matric potentials. The data of Willaredt and Nehls (2021) was represented with the PDI (Peters, 2013; Iden and Durner, 2014; Peters, 2014) model with the unconstrained bimodal (Durner, 1994) basic function of van Genuchten (1980) and the data set of Sakaki and Smits (2015) was described with the PDI model using the constrained bimodal van Genuchten function (Durner, 1994). The magnitude reflects the differences between the replicates due to different sampling strategies (packing cylinders to a defined weight for compaction vs. in situ sampling from containers)."

*Table 4 and Figure 9: Please check the values of log(K) for pF=0 shown in Figure 9 in comparison to the values of Ks presented in Table 4. For example C0E10 (data of Deeb et al. (2016)), log(46)= 1.66. However, this is not the value found in Figure 9 for pF=0 .*

We thank the reviewer for this remark and corrected the header of the table accordingly: $K_s$ is changed to $K_{s,matrix}$, which is the so-called saturated matrix conductivity corresponding to the conductivity at a matric potential of pF 0.8 (as described in the paragraph above Table 4).

*Table caption of Figure 9: Typo: "the the".*

We corrected it accordingly.

*l. 373: "of" is missing. One should read "e.g. in the form of a database".*

We corrected it accordingly.

*Equation A1: "thetas" and "thetar" are not defined.*

Thank your for this remark. We completed the definition as follows:

> "[…] where $\theta$ (h) $[m^3m^{-3}]$ stands for the volumetric water content, h [cm] stands for the matric potential, $\theta_s$ $[m^3m^{-3}]$ stands for the saturated water content and $\theta_r$ $[m^3m^{-3}]$ for the residual water content."

*Table A1 and A2: What do "alpha1", "n1", "n2", "m1", "m2", "w2" mean. The uncertainties obtained for each of the fitted model parameters should be indicated too. This concerns also tables A3 and A4.*

We appreciate this remark of the reviewer and include a description of the meaning of the fitting parameters:

> "[...] the numbers in the subscript indicate the sub function to which the parameters belong."

Regarding the indication of uncertainties for each model parameter we refer the reviewer to our reply to RC1#5.